# GETTING *free* BITS BACK FROM ROTATIONAL SYMMETRIES IN LLMS

## ABSTRACT

Current methods for compressing neural network weights, such as decomposition, pruning, quantization, and channel simulation, often overlook the inherent symmetries within these networks and thus waste bits on encoding redundant information. In this paper, we propose a format based on bits-back coding for storing rotationally symmetric Transformer weights more efficiently than the usual array layout at the same floating-point precision. We evaluate our method on Large Language Models (LLMs) pruned by SliceGPT (Ashkboos et al., 2024) and achieve a 3-5% reduction in total bit usage *for free* across different model sizes and architectures without impacting model performance within a certain numerical precision.

## 1 INTRODUCTION

Modern neural networks, particularly Large Language Models (LLMs), typically contain billions of parameters. Therefore, encoding and transmitting these models efficiently is gaining widespread interest. Currently, compression techniques of model weights mainly fall into four categories, including decomposition (e.g., Hu et al., 2022; Saha et al., 2023), pruning (e.g., Hoefler et al., 2021; Frantar & Alistarh, 2023; Ashkboos et al., 2024), quantization (e.g., Wang et al., 2023; Xu et al., 2024), and channel simulation (e.g., Havasi et al., 2019; Isik et al., 2023; He et al., 2024).

However, these techniques ignore the fact that neural networks typically exhibit symmetries in their weight space. For example, in feedforward networks, applying a random permutation to the neurons in one layer and its inverse to the weights in the subsequent layer leaves the output unchanged. Encoding weights without accounting for these symmetries will lead to suboptimal codelength.

In this work, we address this redundancy by developing a practical storage format for model weights that takes symmetries into account to reduce the compressed model size. We demonstrate the practicality of our method by compressing popular model architectures. Specifically, our contributions are as follows:

- We propose a practical bits-back coding scheme for rotational symmetries. We apply our approach to Large Language Models (LLMs) pruned by SliceGPT (Ashkboos et al., 2024) and demonstrate that our proposed approach can save additional free bits while preserving prediction accuracy within a certain numerical precision.

- We further showcase that by transmitting a small number of bits as a correction code, we can rescue the performance drops due to numerical inaccuracies.

- We perform experiments on the OPT (Zhang et al., 2022) and Llama-2 (Touvron et al., 2023) across different sizes, where we can save 3-5% additional bits *for free*. Notably, our method is completely training-free and can be executed on a consumer-grade GPU or even CPU. Furthermore, our proposed method only adds minimal overhead to the time it takes to load the model parameters into memory and does not affect inference latency.

## 2 BACKGROUND

Before discussing our methods, we provide a brief introduction to bits-back coding (Frey & Hinton, 1996), Transformer (Vaswani et al., 2017), and SliceGPT (Ashkboos et al., 2024).

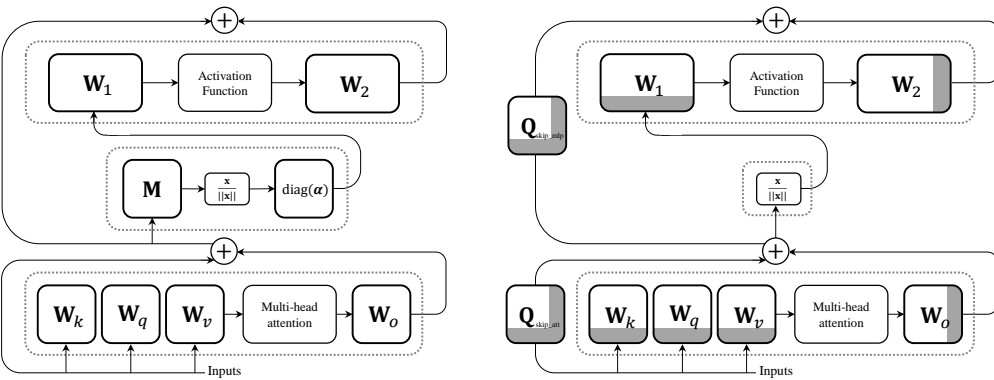

(a) A standard transformer block.    (b) A transformer block with SliceGPT.

Figure 1: Visualization of a Standard Transformer Block and a SliceGPT-Pruned Transformer Block. (a) The standard Transformer block first maps the input through an attention layer; then it applies LayerNorm (Ba et al., 2016) and a 1-layer Feedforward Network (FFN). Two residual connections are added after the attention layer and the FFN. Here, we adopt the notation by Ashkboos et al. (2024), where $\mathbf{M} = \mathbf{I} - \frac{1}{D}\mathbf{1}\mathbf{1}^{\top}$ represents the operation that subtracts the mean in each row. (b) SliceGPT (Ashkboos et al., 2024) first absorbs $\mathbf{M}$ and $\text{diag}(\boldsymbol{\alpha})$ into the weights before and after the normalization layer. It then rotates these weights by applying PCA to the hidden states, aligning them with their principal components (PCs). Subsequently, SliceGPT prunes rows and columns corresponding to the least significant PCs, indicated by gray shadows. It is important to note that the weights in (b) differ from those in (a) due to the absorption of $\mathbf{M}$ and $\text{diag}(\boldsymbol{\alpha})$ and the rotation. Additionally, as SliceGPT introduces two weight matrices $\mathbf{Q}_{\text{skip\_mlp}}$ and $\mathbf{Q}_{\text{skip\_att}}$ to the skip connections, it carries more rotational symmetries compared to the standard Transformer in (a). For a more detailed explanation of SliceGPT, please refer to Figure 4 in Ashkboos et al. (2024).

**Bits-back Coding.**    The motivating idea behind *bits-back coding* (Hinton & Van Camp, 1993; Townsend et al., 2019) can be summarised as follows: "*If we can make multiple equivalent choices to encode something, we should make our choice at random.*" Note that transmitting this random choice requires some bits, and the bits-back coding algorithm provides a concrete procedure to recover the bits we used to randomize our choice. The procedure is based on the following insight from compression: assuming we have the right coding distribution $P$, the encoding function of a compressor will output a sequence of uniformly random bits. Therefore, if we run this process in reverse and run the decoder on a sequence of uniformly random bits, it will output a sample following $P$! Therefore, *lossless de-compression* can be viewed as a computational way of performing inverse transform sampling, which provides an invertible way to make the aforementioned random choice.

To make the bits-back mechanism more precise, assume we have some data $x$ that belongs to some equivalence class $[x]$. In many cases, encoding only the equivalence class $[x]$ instead of a specific instance $x$ would be enough for the task at hand. Given a new item $x$ and a stream of already compressed bits $\mathcal{M}$, bits-back coding uses the decoder of lossless compressor on $\mathcal{M}$ to decode a random element of the equivalence class $x' \sim P_{x|[x]}$ and leaves a shorter message $\mathcal{M}'$. After this, bits-back coding uses the encoder of the compressor to encode $x'$ using $P_x$ as the coding distribution and append it to $\mathcal{M}'$. This procedure is reversible and hence decodable, so long as the receiver of the message can recover $x$ upon seeing $x'$. This ensures that $x'$ can be coded back into the stream to recover the original message $\mathcal{M}$. As one of our contributions, in section 3.2, we explain how such a recovery step can be carried out when $x$ is a weight matrix and $[x]$ is an equivalence class under a certain rotational symmetry.

A concern with bits-back coding is its initialization: we need an initial stream of bits $\mathcal{M}_0$ to encode the first item. While $\mathcal{M}_0$ represents a significant overhead if we only encode a few items, it only causes a constant overhead and quickly becomes negligible as the number of encoded items grows.

**Transformer Architecture and SliceGPT.** Transformer (Vaswani et al., 2017) is the cornerstone of most Large Language Models. Its basic component is the transformer block, as shown in Figure 1a. Each block consists of a multi-head attention layer, a LayerNorm (Ba et al., 2016), and a feedforward network (FFN). Two residual connections are added around the attention layer and FFN.

SliceGPT (Ashkboos et al., 2024) is a recently proposed method for pruning weights in Transformer models. The approach leverages the insight that the outcome of LayerNorm (more precisely, RMSNorm, i.e., $\mathbf{x} \leftarrow \mathbf{x}/||\mathbf{x}||$) is invariant if we apply a rotation to the input and its inverse to the output. This rotation matrix and its inverse can be absorbed into the weights before and after the normalization layer. Therefore, by performing PCA on the hidden states, we can choose rotation matrices that align with the principal components. This allows us to prune the rows and columns corresponding to the less significant eigenvalues in the hidden states, effectively reducing the model's complexity without drastically hurting the performance. We visualize each transformer block after rotation and pruning in Figure 1b. The shadow indices the pruned columns and rows.

## 3 GETTING BITS BACK FROM ROTATION SYMMETRIES

In this section, we describe our method, which is based on the observation of rotational symmetries in the Transformer block pruned by SliceGPT. Comparing Figure 1b and Figure 1a, we can see SliceGPT not only reduces the number of parameters (by pruning out columns and rows), but also introduces rotational symmetries. We should note that these rotational symmetries do not exist in the standard transformer due to the skip connections. Concretely, in a SliceGPT-pruned Transformer, denoting the weights in the $\ell$-th transformer block with superscripts, we have:

**Remark 3.1.** *Outputs remain unchanged if rotating* $\mathbf{W}_2^{(\ell-1)}$, $\mathbf{b}_2^{(\ell-1)}$ *(if any) and* $\mathbf{Q}_{skip\_mlp}^{(\ell-1)}$ *by an arbitrary orthogonal matrix* $\mathbf{Q}$ [1]*, and rotating* $\mathbf{Q}_{skip\_att}^{(\ell)}, \mathbf{W}_{qkv}^{(\ell)} = \left[\mathbf{W}_k^{(\ell)}, \mathbf{W}_q^{(\ell)}, \mathbf{W}_v^{(\ell)}\right]$ *by* $\mathbf{Q}^\top$ *as follows:*

$$\mathbf{W}_2^{(\ell-1)} \leftarrow \mathbf{W}_2^{(\ell-1)}\mathbf{Q}, \quad \mathbf{b}^{(\ell-1)} \leftarrow \mathbf{b}^{(\ell-1)}\mathbf{Q}, \quad \mathbf{Q}_{skip\_mlp}^{(\ell-1)} \leftarrow \mathbf{Q}_{skip\_mlp}^{(\ell-1)}\mathbf{Q} \tag{1}$$

$$\mathbf{Q}_{skip\_att}^{(\ell)} \leftarrow \mathbf{Q}^\top\mathbf{Q}_{skip\_att}^{(\ell)}, \quad \mathbf{W}_{qkv}^{(\ell)} \leftarrow \mathbf{Q}^\top\mathbf{W}_{qkv}^{(\ell)} \tag{2}$$

*Similarly, outputs remain unchanged if rotating* $\mathbf{W}_o^{(\ell)}$, $\mathbf{b}_o^{(\ell)}$ *(if any) and* $\mathbf{Q}_{skip\_att}^{(\ell)}$ *by* $\mathbf{Q}$*, and rotating* $\mathbf{Q}_{skip\_mlp}^{(\ell)}$ *and* $\mathbf{W}_1^{(\ell)}$ *by* $\mathbf{Q}^\top$ *as follows:*

$$\mathbf{W}_o^{(\ell)} \leftarrow \mathbf{W}_o^{(\ell)}\mathbf{Q}, \quad \mathbf{b}_o^{(\ell)} \leftarrow \mathbf{b}_o^{(\ell)}\mathbf{Q}, \quad \mathbf{Q}_{skip\_att}^{(\ell)} \leftarrow \mathbf{Q}_{skip\_att}^{(\ell)}\mathbf{Q} \tag{3}$$

$$\mathbf{Q}_{skip\_mlp}^{(\ell)} \leftarrow \mathbf{Q}^\top\mathbf{Q}_{skip\_mlp}^{(\ell)}, \quad \mathbf{W}_1^{(\ell)} \leftarrow \mathbf{Q}^\top\mathbf{W}_1^{(\ell)} \tag{4}$$

This observation points to an important challenge when encoding the model for storage: we only really wish to encode the function represented by the weights, but we see that infinitely many different weights can represent the same function. In particular, the transformer weights exhibit multiple equivalent representations due to rotational symmetry. Thus, we adapt bits-back coding (Hinton & Van Camp, 1993; Townsend et al., 2019) to this setting, eliminating precisely this redundancy.

We first offer an informal explanation to clarify this redundancy. For simplicity, let's denote the weights in a transformer as $\mathbf{\Theta}$. Assuming the coding distribution is $P$ [2], we need to spend about $-\log_2 P(\mathbf{\Theta})$ bits to encode the weights directly. On the other hand, as discussed above, applying rotations (and its inversion) to some weights leaves the output invariant. Therefore, if we define *equivalence* in terms of outputs (and we do!), the weights with different rotations form an *equivalence class*, denoted by $[\mathbf{\Theta}]$. Encoding this equivalence class will require $-\log_2\left(\sum_{\mathbf{\Theta}\in[\mathbf{\Theta}]} P(\mathbf{\Theta})\right)$ bits. In a finite-precision system, where the number of possible rotation matrices is limited, the equivalence class is finite. Assuming that each entry in the equivalence class has the same probability, and denoting the cardinality of the equivalence class by $\mathcal{C}$, we have $-\log_2\left(\sum_{\mathbf{\Theta}\in[\mathbf{\Theta}]} P(\mathbf{\Theta})\right) = -\log_2(\mathcal{C}P(\mathbf{\Theta})) = -\log_2 P(\mathbf{\Theta}) - \log_2 \mathcal{C}$. This implies that directly encoding the weights wastes $-\log_2 \mathcal{C}$ bits more than necessary.

We apply bits-back coding to eliminate this redundancy. In short, each time we encode the weights in one transformer block (more precisely, $\mathbf{W}_2^{(\ell)}$ and $\mathbf{W}_o^{(\ell)}$), we start by decoding a random rotation

---

[1] Throughout this paper, we will use orange-colored $\mathbf{Q}$ to denote orthogonal matrices.

[2] If we encode the weights using `float16`, we are essentially assuming that all possible floating-point values ($2^{16}$ in total) have the same probability mass.

---

**Algorithm 1** Rotate Transformer to its Canonical Direction.

---

**Input:** Transformer weights with SliceGPT: $\mathbf{W}_{\text{emb}}$, $\mathbf{Q}_{\text{skip\_att}}^{(\ell)}$, $\mathbf{W}_{qkv}^{(\ell)}$, $\mathbf{W}_o^{(\ell)}$, $\mathbf{b}_{qkv}^{(\ell)}$, $\mathbf{b}_o^{(\ell)}$, $\mathbf{Q}_{\text{skip\_mlp}}^{(\ell)}$, $\mathbf{W}_1^{(\ell)}$, $\mathbf{W}_2^{(\ell)}$, $\mathbf{b}_1^{(\ell)}$, $\mathbf{b}_2^{(\ell)}$, $\mathbf{W}_{\text{head}}$, $\mathbf{b}_{\text{head}}$, $\ell = 1, 2, \cdots, L$;
**Output:** Rotated weights.

  # rotate input embeddings:
  $\mathbf{Q} \leftarrow$ Eigenvalue Decompsition($\mathbf{W}_{\text{emb}}^\top \mathbf{W}_{\text{emb}}$);
  $\mathbf{W}_{\text{emb}} \leftarrow \mathbf{W}_{\text{emb}} \mathbf{Q}$;
  **for** $\ell \in [1, \cdots, L]$ **do**
    # rotate skip connection and attention:
    $\mathbf{Q}_{\text{skip\_att}}^{(\ell)} \leftarrow \mathbf{Q}^\top \mathbf{Q}_{\text{skip\_att}}^{(\ell)}$; $\mathbf{W}_{qkv}^{(\ell)} \leftarrow \mathbf{Q}^\top \mathbf{W}_{qkv}^{(\ell)}$;
    # rotate attention output weight:
    $\mathbf{Q} \leftarrow$ Eigenvalue Decompsition($\mathbf{W}_o^{(\ell)^\top} \mathbf{W}_o^{(\ell)}$);
    $\mathbf{W}_o^{(\ell)} \leftarrow \mathbf{W}_o^{(\ell)} \mathbf{Q}$; $\mathbf{b}_o^{(\ell)} \leftarrow \mathbf{Q}^\top \mathbf{b}_o^{(\ell)}$;
    # rotate skip connection and MLP input weight:
    $\mathbf{Q}_{\text{skip\_att}}^{(\ell)} \leftarrow \mathbf{Q}_{\text{skip\_att}}^{(\ell)} \mathbf{Q}$; $\mathbf{Q}_{\text{skip\_mlp}}^{(\ell)} \leftarrow \mathbf{Q}^\top \mathbf{Q}_{\text{skip\_mlp}}^{(\ell)}$; $\mathbf{W}_1^{(\ell)} \leftarrow \mathbf{Q}^\top \mathbf{W}_1^{(\ell)}$;
    # rotate skip connection and MLP output weight:
    $\mathbf{Q} \leftarrow$ Eigenvalue Decompsition($\mathbf{W}_2^{(\ell)^\top} \mathbf{W}_2^{(\ell)}$);
    $\mathbf{Q}_{\text{skip\_mlp}}^{(\ell)} \leftarrow \mathbf{Q}_{\text{skip\_mlp}}^{(\ell)} \mathbf{Q}$; $\mathbf{W}_2^{(\ell)} \leftarrow \mathbf{W}_2^{(\ell)} \mathbf{Q}$; $\mathbf{b}_2^{(\ell)} \leftarrow \mathbf{Q}^\top \mathbf{b}_2^{(\ell)}$;
  **end for**
  # rotate heads:
  $\mathbf{W}_{\text{head}} \leftarrow \mathbf{Q}^\top \mathbf{W}_{\text{head}}$;

---

**Algorithm 2** Recover rotation matrix from rotated weight.

---

**Input:** Rotated matrix $\mathbf{W}$, reference signs $\mathbf{s}$ (vector of $\pm 1$-s).
**Output:** Rotation matrix $\mathbf{Q}$:
  $\mathbf{Q}^\top \leftarrow$ Eigenvalue Decompsition($\mathbf{W}^\top \mathbf{W}$).       ▷ rotate $\mathbf{W}_2^{(\ell)}$ to canonical direction
  **for** $r \in |\text{row}(\mathbf{Q})|$ **do**
    $\mathbf{Q}_r \leftarrow \begin{cases} \mathbf{Q}_r, & \text{if } \text{sign}(\mathbf{Q}_r.\text{sum()}) = \mathbf{s}_r; \\ -\mathbf{Q}_r, & \text{otherwise.} \end{cases}$     ▷ change sign of $\mathbf{Q}$
  **end for**

---

**Algorithm 3** Decode a rotation matrix from the current bitstream. We use **red** to represent adding bits to the bitstream; **green** for removing bits from the bitstream.

---

**Input:** Bitstream $\mathcal{M}$;
**Output:** Rotation matrix $\mathbf{Q} \in \mathbb{R}^{D \times D}$.
  $\mathbf{X} \leftarrow \mathbf{0} \in \mathbb{R}^{D \times D}$;
  Decode $D(D-1)/2$ floats from bitstream $\mathcal{M}$;
  Fill above the diagonal of $\mathbf{X}$ with these floats;
  $\mathbf{X} \leftarrow \mathbf{X} + \mathbf{X}^\top$;
  Decode $D$ floats from bitstream $\mathcal{M}$;
  Fill the diagonal of $\mathbf{X}$ with these floats;
  $\mathbf{Q}, \boldsymbol{\lambda} \leftarrow$ Eigenvalue Decomposition($\mathbf{X}$);
  $\boldsymbol{\lambda} \rightarrow \texttt{Encode\_to}(\mathcal{M})$.

---

**Algorithm 4** Encode a rotation matrix to the current bitstream. We use **red** to represent adding bits to the bitstream; **green** for removing bits from the bitstream.

---

**Input:** Rotation matrix $\mathbf{Q}$, Bitstream $\mathcal{M}$;
**Output:** Updated bitstream $\mathcal{M}$.

  $\boldsymbol{\lambda} \leftarrow \texttt{Decode\_from}(\mathcal{M})$.
  $\mathbf{X} \leftarrow \mathbf{Q} \, \text{diag}(\boldsymbol{\lambda}) \, \mathbf{Q}^\top$.
  Retrieve floats in the diagonal of $\mathbf{X}$;
  Encode these $D$ floats into $\mathcal{M}$.
  Retrieve floats in the upper triangular of $\mathbf{X}$;
  Encode these $D(D-1)/2$ floats into $\mathcal{M}$.

---

from the current bitstream and applying it to the weights. We then encode the rotated weights into the bitstream. When decoding, we first decode the rotated weights and recover the rotation we applied to the original weights. Then, we encode the rotation matrix back to the bitstream. This process is repeated for every transformer block. One concern the reader might have regarding our proposed

method is that bits-back coding is known to have poor one-shot compression performance and is only effective when encoding large datasets. This poor performance is mainly due to the fact that we need some initial bits to perform bits-back, causing overhead that will only be eliminated asymptotically. However, this is not an issue in our approach due to two reasons: (1) in the Transformer, besides the transformer blocks, we also need to store a relatively large head and embedding layer. We can simply use this as the initial bits for bits-back; and (2) note that we apply our coding technique to *each* transformer block in the Transformer. We can view this single Transformer as a dataset consisting of transformer blocks as the elements. For large enough architectures (such as the ones we used in our experiments), the bits-back coding is already efficient.

However, there are two questions that remain unsolved: (a) *How can we recover the rotation given a rotated weight matrix?* (b) *How can we decode/encode a rotation (Orthogonal) matrix from/to the current bitstream?* We will answer these questions in Section 3.1 and Section 3.2, respectively. We then put things all together in Section 3.3 and describe the full encoding and decoding algorithms in Algorithms 5 and 6. Finally, as we only apply rotations to weight matrices with finite precision (e.g., `float16`), we may suffer from numerical inaccuracy, impacting the transformer's outputs. To handle this, we propose to send a simple correction code, which we discuss at the end of Section 3.3.

## 3.1 ROTATING TRANSFORMER WEIGHTS TO THEIR CANONICAL DIRECTION

We now discuss how to recover the rotation from a rotated weight matrix. This is, in general, not feasible without additional information about the original weights. Fortunately, as noted in Remark 3.1, we can apply any rotation to the weights. This allows us to first rotate the weights to a *canonical direction* as a reference. We can define this canonical direction in multiple ways as long as we can recover it easily after applying a random rotation. In this work, we adopt eigenvalue decomposition to define the canonical direction, while future works could explore more sophisticated methods.

We detail the algorithm for the canonical direction in Algorithm 1. In short, for each transformer block, we can apply two free rotations according to Remark 3.1: the first rotation is applied to $\mathbf{W}_2^{(\ell-1)}$, $\mathbf{b}_2^{(\ell-1)}$, $\mathbf{Q}_{\text{skip\_mlp}}^{(\ell-1)}$, $\mathbf{Q}_{\text{skip\_att}}^{(\ell)}$, and $\mathbf{W}_{qkv}^{(\ell)}$. We hence define the canonical direction such that $\mathbf{W}_2^{(\ell-1)}{}^\top \mathbf{W}_2^{(\ell-1)}$ is diagnol; the second rotation is applied to $\mathbf{W}_o^{(\ell)}$, $\mathbf{b}_o^{(\ell)}$, $\mathbf{Q}_{\text{skip\_att}}^{(\ell)}$, $\mathbf{Q}_{\text{skip\_mlp}}^{(\ell)}$ and $\mathbf{W}_1^{(\ell)}$. We hence define the canonical direction such that $\mathbf{W}_o^{(\ell)}{}^\top \mathbf{W}_o^{(\ell)}$ is diagonal.

After rotating the transformer to its canonical direction, we can recover any rotation that is applied to the canonical $\mathbf{W}_2^{(\ell-1)}$ or $\mathbf{W}_o^{(\ell)}$ by eigenvalue decomposition. Specifically, let's consider a random rotation $\mathbf{Q}$ applied to $\mathbf{W}_2^{(\ell-1)}$ in its canonical direction as an example. Denoting the matrix after rotation is $\tilde{\mathbf{W}}_2^{(\ell-1)} \leftarrow \mathbf{W}_2^{(\ell-1)}\mathbf{Q}$, we can perform eigenvalue decomposition on $\tilde{\mathbf{W}}_2^{(\ell-1)}{}^\top \tilde{\mathbf{W}}_2^{(\ell-1)}$, and the rotation matrix $\mathbf{Q}$ can then be recovered by stacking the eigenvectors together in columns. The weight matrix in the canonical direction can be obtained by $\mathbf{W}_2^{(\ell-1)} \leftarrow \tilde{\mathbf{W}}_2^{(\ell-1)}\mathbf{Q}^\top = \mathbf{W}_2^{(\ell-1)}\mathbf{Q}\mathbf{Q}^\top$.

A caveat exists in the above procedure: eigenvalue decomposition can result in eigenvectors with opposite signs. This will lead to undesired results when recovering the canonical weight matrix. We include a detailed explanation in Appendix A. To address this, we encode the sign of the summation of each row of the rotation matrix as side information. This only requires $D$ bits for a $D$-dimensional rotation matrix. After recovering eigenvectors through eigenvalue decomposition, we can use this side information to correct the sign for each eigenvector (i.e., rows in the rotation matrix). Algorithm 2 describes this process.

Another concern arises when $\mathbf{W}_2^{(\ell-1)}{}^\top \mathbf{W}_2^{(\ell-1)}$ (or $\mathbf{W}_o^{(\ell)}{}^\top \mathbf{W}_o^{(\ell)}$) is not full-rank. In such cases, eigenvalue decomposition will not recover the rotation applied to these canonical weights. To address this, we can define the canonical direction by applying eigenvalue decomposition to $\mathbf{B}^\top \mathbf{B}$, where $\mathbf{B}^\top = \left[ \mathbf{W}_2^{(\ell-1)}{}^\top, \mathbf{b}_2^{(\ell-1)}{}^\top, \mathbf{W}_k^{(\ell)}, \mathbf{W}_q^{(\ell)}, \mathbf{W}_v^{(\ell)} \right]$ (or $\mathbf{B}^\top = \left[ \mathbf{W}_o^{(\ell)}{}^\top, \mathbf{b}_o^{(\ell)}{}^\top, \mathbf{W}_1^{(\ell)} \right]$). However, we actually found $\mathbf{W}_2^{(\ell-1)}{}^\top \mathbf{W}_2^{(\ell-1)}$ and $\mathbf{W}_o^{(\ell)}{}^\top \mathbf{W}_o^{(\ell)}$ were already full-rank across all architectures in our experiments. This may be because SliceGPT has already pruned insignificant principal components in the hidden states, leading to more compact weight matrices.

## 3.2 DECODING AND ENCODING ROTATION MATRICES

Now, we discuss *how to decode/encode a rotation matrix from/to a given bitstream.* A naive approach is to directly decode and encode these $D^2$ entries in a rotation matrix $\mathbf{Q} \in \mathbb{R}^{D \times D}$, e.g., by float16. However, it is difficult to guarantee that $D^2$ elements decoded from a given bitstream can form a rotation matrix. In fact, a $D$-dimensional rotation matrix $\mathbf{Q}$ has only $D(D-1)/2$ degrees of freedom (DOF), which means that we only need to decode and encode $D(D-1)/2$ floats for the entire matrix. Therefore, the question becomes: *(a) how can we construct a random rotation matrix from $D(D-1)/2$ random floats; (b) how can we recover these floats given a rotation matrix?*

Ideally, we aim to generate a uniformly distributed random rotation matrix, i.e., a random rotation matrix from the Haar distribution. Following the method by Stewart (1980), we can construct the matrix by iteratively applying Householder transformations (Householder, 1958).

However, this algorithm is difficult to reverse: we need to reverse the householder transformations one by one, and hence, we will suffer from large numerical instability. Therefore, we propose a simple method to generate a rotation matrix. This approach does not result in a uniformly distributed rotation matrix. However, we found our approach works well in practice. Since our goal is not to design a theoretically optimal algorithm but rather a more practical approach to perform bits-back, we leave a better design for the rotation matrix to future works.

We describe the process of decoding and encoding a rotation matrix in Algorithms 3 and 4. Again, we employ a bits-back approach for efficiency. In brief, to decode a rotation matrix, we first decode a symmetric matrix from the bitstream by decoding its diagonal and upper triangular parts and performing an eigenvalue decomposition. The eigenvalues are then encoded back into the bitstream. To encode this rotation matrix, we first decode its eigenvalues from the bitstream, reconstruct the symmetric matrix via matrix multiplication, and then encode its diagonal and upper triangular parts back into the bitstream. Notably, our approach requires only the number of bits corresponding to $D(D-1)/2$ floats, which aligns with the degrees of freedom of a random rotation matrix.

## 3.3 PUTTING THINGS TOGETHER AND HANDLING NUMERICAL INACCURACY

Having discussed the canonical direction for the transformer and the algorithm for decoding and encoding a rotation matrix, we detail the complete algorithm for encoding and decoding the entire transformer using bits-back in Algorithms 5 and 6, respectively. In these algorithms, we use Encode_to and Decode_from to represent the process of appending or popping arrays of float16 values into or from the current bitstream.

However, since we only save rotated weights in finite precision (e.g., float16), we may suffer from numerical inaccuracy, and hence the rotation matrix recovered by Algorithm 2 in decoding will have deviations from the original rotation matrix applied to the canonical weights in encoding. This will lead to two undesirable outcomes: (1) the bitstream after re-encoding the rotation matrices (as shown in lines 7 and 13 in Algorithm 6) will contain errors, which will affect the weights decoded subsequently from this bitstream; (2) the weight matrices rotated back to the canonical direction (as shown in lines 6 and 12 in Algorithm 6) will contain errors.

The first error can be fatal in standard bits-back coding, as they are usually implemented using a *variable-length* code, such as asymmetric numeral systems (Duda, 2009; Townsend et al., 2019). Such a system is very sensitive to decoding errors: as the code assigns different codelengths to symbols by design, if the decoder can only recover the compressed data approximately due to numerical errors, not only are they not getting the correct bits back, they might not even get the correct *number* of bits back. If the decoder makes such an error even once, it misaligns the rest of the bitstream (i.e., it will be longer or shorter than it should be), causing catastrophic decoding errors.

On the other hand, our proposed method is robust to such errors because we implement bits-back coding with a fixed-length code: we set a floating-point precision (e.g., 16 bits) ahead of time. Then, each encoding and decoding operation will change the message length by the same amount: for a fixed precision, we can compute the total codelength of the model ahead of time. Importantly, this means that any decoding error remains local: if we do not recover a given weight $w$ exactly, this will only affect the value of $w$ but will not affect the rest of the bitstream.

---

**Algorithm 5** Bits-back Encoding for transformers (processed by SliceGPT). We use red to represent adding bits to the bitstream; green to represent removing bits from the bitstream.

---

**Input:** Transformer weights: $\mathbf{W}_{\text{emb}}, \mathbf{Q}_{\text{skip\_att}}^{(\ell)}, \mathbf{W}_{qkv}^{(\ell)}, \mathbf{W}_o^{(\ell)}, \mathbf{b}_{qkv}^{(\ell)}, \mathbf{b}_o^{(\ell)}, \mathbf{Q}_{\text{skip\_mlp}}^{(\ell)}, \mathbf{W}_1^{(\ell)}, \mathbf{W}_2^{(\ell)}, \mathbf{b}_1^{(\ell)},$
     $\mathbf{b}_2^{(\ell)}, \mathbf{W}_{\text{head}}, \mathbf{b}_{\text{head}}, \ell = 1, 2, \cdots, L;$
**Output:** Binary message $\mathcal{M}$.

1: $\mathcal{M} \leftarrow \perp.$        ▷ initialization empty bitstream.
2: Rotate the transformer to its canonical direction using Algorithm 1.
3: # encode weights with bits-back:
4: $\mathbf{W}_{\text{emb}}, \mathbf{b}_{\text{emb}} \rightarrow \texttt{Encode\_to(}\mathcal{M}\texttt{)}.$      ▷ encode input embeddings
5: **for** $\ell \in [1, \cdots, L]$ **do**
6:      $\mathbf{Q}_{\text{skip\_att}}^{(\ell)}, \mathbf{W}_{qkv}^{(\ell)}, \mathbf{b}_{qkv}^{(\ell)} \rightarrow \texttt{Encode\_to(}\mathcal{M}\texttt{)}.$
7:      $\mathbf{Q} \leftarrow$ Decode rotation matrix from $\mathcal{M}$ using Algorithm 3.    ▷ decode a random rotation
8:      $\mathbf{W}_o^{(\ell)} \leftarrow \mathbf{W}_o^{(\ell)} \mathbf{Q}.$
9:      $\texttt{sign(}\mathbf{Q}\texttt{.sum(-1))} \rightarrow \texttt{Encode\_to(}\mathcal{M}\texttt{)}.$      ▷ encode sign of $\mathbf{Q}$ (overhead)
10:      $\mathbf{W}_o^{(\ell)}, \mathbf{b}_o^{(\ell)}, \mathbf{Q}_{\text{skip\_mlp}}^{(\ell)}, \mathbf{W}_1^{(\ell)}, \mathbf{b}_1^{(\ell)} \rightarrow \texttt{Encode\_to(}\mathcal{M}\texttt{)}.$
11:           ▷ encode rotated $\mathbf{W}_o^{(\ell)}$ and other weights
12:      $\mathbf{Q} \leftarrow$ Decode rotation matrix from $\mathcal{M}$ using Algorithm 3.    ▷ decode a random rotation
13:      $\mathbf{W}_2^{(\ell)} \leftarrow \mathbf{W}_2^{(\ell)} \mathbf{Q}.$
14:      $\texttt{sign(}\mathbf{Q}\texttt{.sum(-1))} \rightarrow \texttt{Encode\_to(}\mathcal{M}\texttt{)}.$      ▷ encode sign of $\mathbf{Q}$ (overhead)
15:      $\mathbf{W}_2^{(\ell)}, \mathbf{b}_2^{(\ell)} \rightarrow \texttt{Encode\_to(}\mathcal{M}\texttt{)}.$    ▷ encode rotated $\mathbf{W}_2^{(\ell)}$ and other weights
16: **end for**
17: $\mathbf{W}_{\text{head}}, \mathbf{b}_{\text{head}} \rightarrow \texttt{Encode\_to(}\mathcal{M}\texttt{)}.$      ▷ encode heads

---

**Algorithm 6** Bits-back Decoding for transformers (processed by SliceGPT). We use red to represent adding bits to the bitstream; green to represent removing bits from the bitstream.

---

**Input:** Binary message $\mathcal{M}$.
**Output:** Transformer weights: $\mathbf{W}_{\text{emb}}, \mathbf{Q}_{\text{skip\_att}}^{(\ell)}, \mathbf{W}_{qkv}^{(\ell)}, \mathbf{W}_o^{(\ell)}, \mathbf{b}_{qkv}^{(\ell)}, \mathbf{b}_o^{(\ell)}, \mathbf{Q}_{\text{skip\_mlp}}^{(\ell)}, \mathbf{W}_1^{(\ell)}, \mathbf{W}_2^{(\ell)},$
     $\mathbf{b}_1^{(\ell)}, \mathbf{b}_2^{(\ell)}, \mathbf{W}_{\text{head}}, \mathbf{b}_{\text{head}}, \ell = 1, 2, \cdots, L.$

1: $\mathbf{W}_{\text{head}}, \mathbf{b}_{\text{head}} \leftarrow \texttt{Decode\_from(}\mathcal{M}\texttt{)}.$      ▷ decode heads
2: **for** $\ell \in [L, \cdots, 1]$ **do**
3:      $\mathbf{W}_2^{(\ell)}, \mathbf{b}_2^{(\ell)} \leftarrow \texttt{Decode\_from(}\mathcal{M}\texttt{)}.$    ▷ decode rotated $\mathbf{W}_2^{(\ell)}$ and other weights
4:      $\mathbf{s} \leftarrow \texttt{Decode\_from(}\mathcal{M}\texttt{)}.$      ▷ decode sign of $\mathbf{Q}$
5:      $\mathbf{Q} \leftarrow$ Recover rotation matrix using Algorithm 2 from $(\mathbf{W}_2^{(\ell)}, \mathbf{s}).$
6:      $\mathbf{W}_2^{(\ell)} \leftarrow \mathbf{W}_2^{(\ell)} \mathbf{Q}^{\top}$      ▷ recover canonical direction
7:      $\mathbf{Q} \rightarrow$ Encode rotation matrix to $\mathcal{M}$ using Algorithm 4.    ▷ encode the random rotation
8:      $\mathbf{W}_o^{(\ell)}, \mathbf{b}_o^{(\ell)}, \mathbf{Q}_{\text{skip\_mlp}}^{(\ell)}, \mathbf{W}_1^{(\ell)}, \mathbf{b}_1^{(\ell)} \leftarrow \texttt{Decode\_from(}\mathcal{M}\texttt{)}.$
9:           ▷ decode rotated $\mathbf{W}_o^{(\ell)}$ and other weights
10:      $\mathbf{s} \leftarrow \texttt{Decode\_from(}\mathcal{M}\texttt{)}.$      ▷ decode sign of $\mathbf{Q}$
11:      $\mathbf{Q} \leftarrow$ Recover rotation matrix using Algorithm 2 from $(\mathbf{W}_o^{(\ell)}, \mathbf{s}).$
12:      $\mathbf{W}_o^{(\ell)} \leftarrow \mathbf{W}_o^{(\ell)} \mathbf{Q}^{\top}$      ▷ recover canonical direction
13:      $\mathbf{Q} \rightarrow$ Encode rotation matrix to $\mathcal{M}$ using Algorithm 4.    ▷ encode the random rotation
14:      $\mathbf{Q}_{\text{skip\_att}}^{(\ell)}, \mathbf{W}_{qkv}^{(\ell)}, \mathbf{b}_{qkv}^{(\ell)} \leftarrow \texttt{Decode\_from(}\mathcal{M}\texttt{)}.$
15: **end for**
16: $\mathbf{W}_{\text{emb}}, \mathbf{b}_{\text{emb}} \leftarrow \texttt{Decode\_from(}\mathcal{M}\texttt{)}.$      ▷ decode input embeddings

---

However, although our bits-back process will not propagate local decoding errors, individual errors itself can still impact the model performance. Therefore, we propose transmitting an additional correction code to correct errors exceeding a certain threshold. Specifically, errors in (a) occur in the

$D(D+1)/2$ floats obtained by Algorithm 3 when encoding the rotation matrix to the bitstream, and errors in (b) occur when rotating the weight matrices back to the canonical direction. Note that the encoder can simulate both procedures during encoding to determine the exact value that the decoder will obtain. If the error between the value obtained by the decoder and the one held by the encoder exceeds a certain threshold, the encoder can send a correction code containing the positions and the true values in `float16`. Correcting each value will require approximately $16 + \lceil \log_2 L \rceil$ bits, where $L$ is the total number of values the decoder will reconstruct that can have errors. For example, $L = D(D+1)/2$ for the error caused by (a), and $L$ represents the total number of parameters in the weight matrix for the error caused by (b).

A natural concern is that the correction code could become large if there are too many errors. Fortunately, as we show in Figure 2, only a tiny portion of values have relatively large errors. Therefore, the correction code requires only a small number of bits to transmit and does not significantly impact the overall coding efficiency. It is worth noting that this correcting strategy can be considered a simple error-correction code. Therefore, we may be able to adopt more complex error-correction codes, but we leave this design for future exploration.

### 3.4 Analysis of the Codelength

Here, we analyze the codelength reduction achieved by our proposed approach from a practical standpoint. A more rigorous theoretical analysis is provided in Appendix B. For simplicity's sake, we assume there is no bias vector in our transformer architecture. This is a reasonable assumption, as some modern architectures like Llama (Touvron et al., 2023) omit the bias too. Additionally, we assume the transformer has no output head or embedding layer. This assumption can be interpreted as modeling an extremely deep transformer, where the effects of the head and embedding layers become negligible. However, it is important to note that this is not a realistic assumption in practical scenarios. This is the main reason for the discrepancy between our analysis in this section and the results we present in Section 4.

In one transformer block, as shown in Figure 1b, there exist eight matrices after SliceGPT, including six sliced weight matrices and two skip connection matrices. If the slicing rate is $s$ and the weights are stored at $\delta$ bits precision (for example, $\delta = 16$ in `float16`), the total codelength (in bits) can be expressed as:

$$( \underbrace{6 \cdot rD^2}_{\text{6 weight matrices}} + \underbrace{2 \cdot (rD)^2}_{\text{2 skip connection}} ) \cdot \delta \tag{5}$$

where we denote $r = 1 - s$ as the remaining rate after slicing. Using bits-back, we decode two rotation matrices from the bitstream during encoding, leading to a reduction in codelength by:

$$\left( 2 \cdot \frac{(rD)(rD-1)}{2} \right) \cdot \delta = (rD) \cdot (rD - 1) \cdot \delta \tag{6}$$

We disregard the overhead from storing the signs of the eigenvectors (line 9 in Algorithm 5) and the correction codes (discussed in Section 3.3), as these contributions are negligible.

Thus, the overall reduction in codelength is:

$$(rD) \cdot (rD - 1)/(6 \cdot rD^2 + 2 \cdot (rD)^2) \approx r/(6 + 2r) \tag{7}$$

For a slice rate of $s = 20 - 30\%$, this results in approximately a $10\%$ reduction in codelength.

## 4 Experiments and Results

We evaluate our proposed approach in this section. We first test our method on the Open Pre-trained Transformer Language Models (OPT, Zhang et al., 2022) and Llama-2 (Touvron et al., 2023) pruned by SliceGPT (Ashkboos et al., 2024) with different slicing rates. Then, we investigate the effectiveness of the correction codes proposed in Section 3.3. We conduct our bits-back algorithms on AMD Ryzen 9 7950X CPU and evaluate the performance on one NVIDIA RTX 4090 GPU.

**Compression rate and performances.** We evaluate our method on OPT-1.3B/2.7B/6.7B/13B and Llama-2-7B, pruned by SliceGPT with different slicing rates in Table 1. We report perplexity (PPL)

Table 1: Compression rates and prediction performances before and after our proposed method. The Compress Rate after SliceGPT takes the correction code's codelength into account. We can see our method reduces further 3-5% bits and has very minor influence on the performance.

| Model | SliceGPT Slicing | Compress Rate after SliceGPT | Compress Rate after bits-back | Performance (before/after bits-back) | | | |
|---|---|---|---|---|---|---|---|
| | | | | PPL (↓) | PIQA (%, ↑) | WinoGrande (%, ↑) | HellaSwag (%, ↑) |
| OPT-1.3B | 20% | -9.53% | -13.77% | **16.59**/16.60 | **64.91**/64.80 | **54.78**/54.38 | 45.26/**45.32** |
| | 25% | -14.84% | -18.61% | **17.78**/17.86 | **63.55**/63.33 | 52.80/**53.28** | **43.20**/43.11 |
| | 30% | -20.53% | -23.81% | **19.60**/19.66 | **60.88**/60.50 | 52.88/**53.28** | **40.25**/40.06 |
| OPT-2.7B | 20% | -9.19% | -13.84% | **13.89**/13.95 | **68.44**/68.12 | **58.88**/58.72 | **51.35**/51.17 |
| | 25% | -15.07% | -19.09% | **14.85**/14.87 | **66.70**/66.76 | 57.30/**57.70** | **48.41**/48.38 |
| | 30% | -20.88% | -24.43% | **16.31**/16.33 | 64.64/**64.69** | 55.80/**56.04** | 44.52/**44.57** |
| OPT-6.7B | 20% | -9.29% | -14.07% | **11.63**/11.71 | 72.91/**73.01** | **61.33**/61.17 | 60.53/**60.55** |
| | 25% | -15.16% | -19.29% | **12.12**/12.15 | 71.00/**71.22** | 60.30/**60.77** | **57.76**/57.55 |
| | 30% | -21.18% | -24.84% | **12.81**/12.91 | 69.31/**69.42** | **59.75**/59.59 | **53.64**/52.94 |
| OPT-13B | 20% | -9.18% | -14.01% | **10.75**/10.77 | 74.27/74.27 | **64.96**/64.88 | 65.74/**65.79** |
| | 25% | -15.27% | -19.51% | 11.08/**11.07** | **74.27**/73.72 | 63.46/**63.93** | **63.48**/63.09 |
| | 30% | -21.29% | -24.97% | **11.55**/11.59 | 72.69/**73.01** | 61.96/**62.43** | **60.12**/60.05 |
| Llama-2-7B | 20% | -9.38% | -14.13% | **6.86**/6.98 | **69.53**/69.42 | 64.17/**64.72** | **58.96**/58.89 |
| | 25% | -15.34% | -19.53% | **7.56**/7.59 | 67.03/**67.57** | 62.98/**63.38** | **54.29**/53.93 |
| | 30% | -21.45% | -25.09% | **8.63**/8.69 | **64.69**/64.09 | **62.75**/62.12 | **49.13**/49.07 |

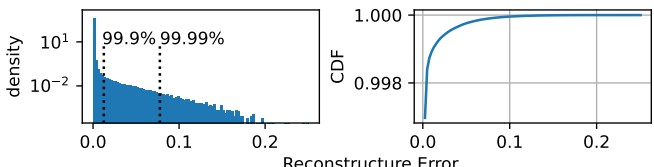 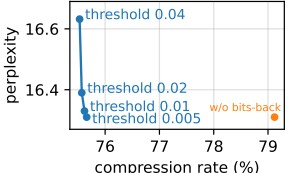

Figure 2: Histogram and empirical CDF of the error between the reconstructed weights and the original weights before encoding, using $\mathbf{W}_o$ in the final layer of OPT-6.7B as an example. The pattern in this plot generalizes well to other weights and models. As shown, only a small fraction of the weights exhibit relatively large deviations. Therefore, we can allocate a negligible number of bits to transmit the positions and true values of these weights, effectively correcting the error caused by numerical inaccuracies.

Figure 3: The effectiveness of the correction codes with different thresholds. Setting a threshold around 0.005-0.01 can effectively rescue all performance drops due to numerical inaccuracies while still significantly reducing bits compared to the compression rate without bits-back.

and accuracy on three downstream tasks (PIQA, Bisk et al. (2020); WinoGrande, Sakaguchi et al. (2021); and HellaSwag, Zellers et al. (2019)) to assess our method's impact on performance. Our approach saves an additional 3-5% in bits with negligible impact on performance. Notably, the performance changes are inconsistent, with occasional improvements after bits-back, suggesting that the changes in the performance are more likely due to randomness than a clear degradation. We also note that this codelength reduction is smaller than the theoretical estimates provided in Section 3.4. The primary reason for this discrepancy is that our analysis does not account for the substantial size of the head and embedding layers.

**Numerical inaccuracy and the effectiveness of the correction codes.** We now examine the impact of numerical inaccuracies and the effectiveness of correction codes proposed in Section 3.3. To provide an intuitive understanding of the numerical issue, we use the weights matrix $\mathbf{W}_o$ from the last layer of OPT-6.7B as an example and visualize the error between the reconstructed weights and the original weights in Figure 2. As we can see, only a tiny fraction of the weights exhibit relatively large errors. Therefore, we can transmit the positions and true values of weights whose deviations exceed a certain threshold, using negligible bits to correct the numerical error.

The threshold is a hyperparameter that balances the codelength and accuracy. In Figure 3, we examine the impact of threshold selection using the OPT-2.7B model. Setting a relatively small threshold (0.005-0.01) effectively mitigates nearly all performance drops due to numerical inaccuracies, while still providing a significant reduction compared to the compression rate without bits-back coding. In our experiments, we use a threshold of 0.01 for OPT models and 0.005 for Llama models.

Table 2: Encoding and decoding time on GPU for different models with varying slicing rates. This includes the time for weight transfers between CPU and GPU. Therefore, we can view this time as the total increase in model saving and loading time introduced by our proposed method.

| Model Name | OPT-1.3B | | OPT-2.7B | | OPT-6.7B | | OPT-13B | |
|---|---|---|---|---|---|---|---|---|
| Slicing | 20% | 30% | 20% | 30% | 20% | 30% | 20% | 30% |
| Encoding time | 15 s | 13 s | 30 s | 24 s | 2.5 min | 1.7 min | 6.5 min | 4.1 min |
| Decoding time | 6 s | 5 s | 14 s | 11 s | 1.2 min | 45 s | 2.5 min | 2 min |

**Runtime Analysis.** We measure the encoding and decoding time on one NVIDIA RTX 4090 GPU in Table 2, and we include the results measured on the CPU in Appendix D.1. Our approach aims to reduce storage space and transmission costs. Therefore, we employ our encoding/decoding algorithm only during model saving/loading. Once the model is decoded and loaded into memory, the inference time is identical to that of SliceGPT.

It is also possible to parallelize the encoding and decoding of individual layers for further acceleration. While we do not present experimental results for this optimization in this paper, we outline its implementation below. We note that standard bits-back coding typically cannot support parallelization because it relies on decoding variable-length bits from a bitstream from previous samples. In our approach, however, two key features enable parallelization: (1) LLMs often have a large embedding layer that provides a sufficiently long bitstream to decode several random rotation matrices. (2) The rotation matrix size is fixed, ensuring that the length of decoded bits for each layer is predetermined. If the embedding layer cannot provide enough bitstream to decode random rotation matrices for all layers, we can divide the layers into multiple groups and encode the layers within each group in parallel. We then encode and decode each group in parallel, using bits from the previous group of layers or the embedding layer for the first group.

## 5    CONCLUSION, LIMITATIONS AND FUTURE DIRECTIONS

In this work, we introduce bits-back coding to encode Large Language Models pruned with SliceGPT. Our approach can save 3-5% additional bits almost *for free* across several different architectures and sizes. While bits-back coding has long been applied in data compression, its application to neural networks, where redundancy and symmetry are prevalent, has been underexplored. Our work attempts to bridge this gap, opening a new direction for model compression. A key takeaway is that by re-parameterizing and pre-processing network weights to explicitly capture symmetries, as demonstrated in SliceGPT, we can leverage bits-back coding to eliminate redundant bits.

Future research can focus on designing improved algorithms for encoding and decoding the random rotation matrix, developing better error-correction codes to manage large deviations due to numerical instability, and integrating our method with other model compression techniques, such as the extremely quantized networks (Ma et al., 2024). Our method's major concern is the numerical instability. While we discuss reducing large deviations by a small number of bits as a correction code, the challenge of making this approach efficient for extremely quantized networks remains open.

Finally, we note that our proposed approach is not specifically limited to Transformer with SliceGPT. The concept of bits-back coding applied to neural networks is general, and we can extend it to many architecture that exhibits symmetries. For instance, in Low-rank Adaptation (LoRA; Hu et al., 2022), the modulation is decomposed as $\mathbf{W} = \mathbf{AB}$. In this setup, applying a rotation $\mathbf{Q}$ to $\mathbf{A}$ as $\mathbf{AQ}$ and to $\mathbf{B}$ as $\mathbf{Q}^\top \mathbf{B}$ preserves $\mathbf{W}$. This allows our approach to be seamlessly integrated into such settings. Another special case of our method is dealing with the permutation invariance. Any MLP with a single, $d$-dimensional hidden layer and activations $\phi$, defined as $f(\mathbf{x} \mid \mathbf{A}, \mathbf{B}) = \mathbf{B}\,\phi(\mathbf{Ax})$ exhibits the following permutation invariance. For a $d \times d$ permutation matrix $\mathbf{P}$, we have $f(\mathbf{x} \mid \mathbf{A}, \mathbf{B}) = f(\mathbf{x} \mid \mathbf{PA}, \mathbf{BP}^{-1})$. However, it is a standard fact that permutation matrices are orthogonal. Hence, our proposed method could be applied to a wide range of network architectures to eliminate the redundancy introduced by the permutation symmetry of the hidden units. Thus, an interesting future direction is to investigate if our method could improve the transmission and storage costs of smaller models, such as ones deployed on edge devices.

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

## A   WHY WE NEED TO ENCODE THE SIGN OF EACH EIGENVECTOR?

First, assume we apply a random rotation matrix $\mathbf{Q}$ to some canonical weight matrix $\mathbf{W}$, and obtain $\tilde{\mathbf{W}} \leftarrow \mathbf{W}\mathbf{Q}$. We can write this rotation matrix as a stack of orthonormal vectors:

$$\mathbf{Q} = \begin{bmatrix} — & \mathbf{q}_1^\top & — \\ & \cdots & \\ — & \mathbf{q}_D^\top & — \end{bmatrix} \tag{8}$$

When we recover the canonical weight matrix, we apply eigenvalue decomposition to $\tilde{\mathbf{W}}^\top\tilde{\mathbf{W}}$. This is possible as $\mathbf{W}^\top\mathbf{W}$ is defined to be diagonal. Therefore, $\mathbf{Q}$ is one solution of eigenvalue decomposition:

$$\tilde{\mathbf{W}}^\top\tilde{\mathbf{W}} = \mathbf{Q}^\top\mathbf{W}^\top\mathbf{W}\mathbf{Q} = \mathbf{Q}^\top\mathbf{\Lambda}\mathbf{Q} \tag{9}$$

However, the solution is not unique. We can write

$$\mathbf{Q}^\top\mathbf{\Lambda}\mathbf{Q} = \begin{bmatrix} | & \cdot & | \\ \mathbf{q}_1 & \cdot & \mathbf{q}_D \\ | & \cdot & | \end{bmatrix} \mathbf{\Lambda} \begin{bmatrix} — & \mathbf{q}_1^\top & — \\ \cdot & \cdot & \cdot \\ — & \mathbf{q}_D^\top & — \end{bmatrix} = \sum_d \lambda_d \mathbf{q}_d \mathbf{q}_d^\top \tag{10}$$

Changing the sign of any $\mathbf{q}_d$ will not influence the results of its outer product. Therefore, we can change the sign of each $\mathbf{q}_d$, and this will still be a valid solution to the eigenvalue decomposition. As an example, WLG, assume by eigenvalue decomposition, we obtain

$$\mathbf{Q}' = \begin{bmatrix} — & -\mathbf{q}_1^\top & — \\ — & \mathbf{q}_2^\top & — \\ & \cdots & \\ — & \mathbf{q}_D^\top & — \end{bmatrix} \tag{11}$$

We recover canonical weight matrix by

$$\tilde{\mathbf{W}}\mathbf{Q}'^\top = \mathbf{W}\mathbf{Q}\mathbf{Q}'^\top = \mathbf{W} \begin{bmatrix} — & \mathbf{q}_1^\top & — \\ & \cdots & \\ — & \mathbf{q}_D^\top & — \end{bmatrix} \begin{bmatrix} | & \cdot & | \\ -\mathbf{q}_1 & \cdot & \mathbf{q}_D \\ | & \cdot & | \end{bmatrix} = \mathbf{W} \begin{bmatrix} -1 & & & \\ & 1 & & \\ & & \ddots & \\ & & & 1 \end{bmatrix} \neq \mathbf{W} \tag{12}$$

Therefore, if we do not control the sign of each eigenvector. We cannot recover the original canonical weight matrix.

## B   BITS-BACK JUSTIFICATION

In this section, we justify our scheme by showing that it can be viewed as a particular instantiation of a bits-back scheme (Townsend et al., 2019; Kunze et al., 2024) with a particular discretization of the probability densities involved. We will first explain why bits-back coding is applicable to networks with rotational invariants in a formal manner. Following that, we will calculate the bits saved through bits-back coding in a more rigorous way. For the sake of generality, we will perform singular value decomposition (SVD) on the weight matrix in this section, which, is equivalent to the eigenvalue decomposition we described in the main text.

Let $\mathbf{W}$ be a $\mathbb{R}^{n \times m}$ real-valued matrix, without loss of generality assume that $n \leqslant m$. Then, we can always write $\mathbf{W}$ via its singular value decomposition (SVD):

$$\mathbf{W} = \mathbf{U}\mathbf{\Sigma}\mathbf{V}^\top, \tag{13}$$

where $\mathbf{U}$ is a $\mathbb{R}^{n \times n}$ orthogonal matrix, $\mathbf{\Sigma}$ is a $\mathbb{R}^{n \times n}$ diagonal matrix and $\mathbf{V}$ is a $\mathbb{R}^{m \times n}$ orthogonal matrix. For brevity, we can write $\mathbf{B} = \mathbf{\Sigma}\mathbf{V}^\top$, and thus we have that any $n \times m$ matrix $\mathbf{W}$ can be written as

$$\mathbf{W} = \mathbf{U}\mathbf{B}. \tag{14}$$

Now, we will say that two matrices $\mathbf{A}, \mathbf{B}$ over the same space are rotationally equivalent $\mathbf{A} \sim \mathbf{B}$ if there exists an orthogonal matrix $\mathbf{Q}$ such that $\mathbf{A} = \mathbf{Q}\mathbf{B}$; denote the equivalence class of $\mathbf{A}$ as $[\mathbf{B}]$.

Now, assume that $\mathbf{B} \sim P_{\mathbf{B}}$ and let $P_{\mathbf{W}|\mathbf{B}}(\mathbf{W}) \propto \mathbb{1}\{\mathbf{W} \in [\mathbf{B}]\}$ be the uniform distribution on $[\mathbf{B}]$, i.e. for an orthogonal matrix $\mathbf{Q}$ we have $P_{\mathbf{W}|\mathbf{B}}(\mathbf{Q}\mathbf{W}) = P_{\mathbf{W}|\mathbf{B}}(\mathbf{W})$. Letting $f^{\mathbf{B}}(\mathbf{Q}) = \mathbf{Q}\mathbf{B}$, this actually shows that $P_{\mathbf{W}|\mathbf{B}}(\mathbf{W}) = f^{\mathbf{B}} \# \mathrm{Unif}(\mathcal{O}(n))$, where $\mathcal{O}(n)$ denotes the $n$-dimensional real orthogonal group, $\#$ denotes a pushforward measure, and $\mathrm{Unif}(\mathcal{O}(n))$ is the Haar measure on $\mathcal{O}(n)$. Note, that this immediately implies that the marginal is also rotationally invariant:

$$P_{\mathbf{W}}(\mathbf{Q}\mathbf{W}) = \int P_{\mathbf{W}|\mathbf{B}}(\mathbf{Q}\mathbf{W} \mid \mathbf{B}) \, dP_{\mathbf{B}}(\mathbf{B}) = \int P_{\mathbf{W}|\mathbf{B}}(\mathbf{W} \mid \mathbf{B}) \, dP_{\mathbf{B}}(\mathbf{B}) = P_{\mathbf{W}}(\mathbf{W}). \tag{15}$$

Now, if the neural network we wish to encode is rotationally invariant, then we can always "standardize" $\mathbf{W}$ first by computing its SVD $\mathbf{W} = \mathbf{U}\mathbf{B}$ and setting $\mathbf{W} \leftarrow \mathbf{B}$. Then, to encode $\mathbf{B}$, we sample a random rotation $\mathbf{Q}$, and encode Importantly, we can always recover $\mathbf{B}$ (up to the signs of the rows of $\mathbf{V}$) by performing an SVD. Therefore, we have the following procedure:

**Before encoding:**

1. Run training algorithm to get $\mathbf{W}$ for a rotationally invariant NN.
2. Compute the SVD $\mathbf{W} = \mathbf{U}\mathbf{B}$, where $\mathbf{B} = \boldsymbol{\Sigma}\mathbf{V}^\top$.
3. Set $\mathbf{W} \leftarrow \mathbf{B}$; this doesn't change the NN output.

**During encoding:**

1. Decode an orthogonal matrix $\mathbf{Q} \sim \mathrm{Unif}(\mathcal{O}(n))$ from the message.
2. Encode $\mathbf{W}' = \mathbf{Q}\mathbf{B}$ using $P_{\mathbf{W}}$ into the message.
3. Compute the SVD of $\mathbf{W}' = \mathbf{Q}\mathbf{B} = \mathbf{Q}'\mathbf{B}'$ and record the $n$ signs of $\mathbf{Q}'$ relative to $\mathbf{Q}$. Concretely, compute the diagonal sign matrix $\sigma$ such that $\sigma\mathbf{Q}' = \mathbf{Q}$. Then, $\sigma$ can be encoded using $n$ bits, one for each sign on the diagonal.

**During decoding**

1. Decode $\sigma$ and $\mathbf{W}'$ using $P_{\mathbf{W}}$.
2. Compute the SVD of $\mathbf{W}' = \mathbf{Q}'\mathbf{B}'$, use $\mathbf{W}'$ (or $\mathbf{B}'$) in the NN.
3. Compute $\mathbf{Q} = \sigma\mathbf{Q}'$.
4. Code $\mathbf{Q}$ back into the stream using $\mathrm{Unif}(\mathcal{O}(n))$.

**Computing the coding cost.** Since $\mathbf{W}$ is continuous, let $p_{\mathbf{W}|\mathbf{B}}$ and and $p_{\mathbf{W}}$ denote the densities of $P_{\mathbf{W}|\mathbf{B}}$ and $P_{\mathbf{W}}$, respectively. Since we cannot encode continuous variables, we now make two approximations. First, we discretize the densities: we fix a precision $\delta$ bits, so given that $\mathbf{W}$ is $n \times m$-dimensional, this gives us a set $\mathcal{W}$ of $2^{nm\delta}$ values we can represent. For a representable matrix $w \in \mathcal{W}$, we set $\hat{P}_{\mathbf{W}}(w) \approx p_{\mathbf{W}}(w) \cdot 2^{-nm\delta}$ and $\hat{P}_{\mathbf{W}|B}(w \mid \mathbf{B}) \approx p_{\mathbf{W}|\mathbf{B}}(w \mid \mathbf{B}) \cdot 2^{-nm\delta}$. These approximations are accurate when the densities are piecewise constant, which is true in this case as $p_{\mathbf{W}|\mathbf{B}}$ is constant by definition, and we shall assume in a moment that $p_{\mathbf{W}}$ is constant as well.

Concretely by our earlier definition, $p_{\mathbf{W}|\mathbf{B}}(w \mid \mathbf{B}) \propto \mathbb{1}[w \in [\mathbf{B}]]$. However, note that since $[\mathbf{B}]$ is a proper *subspace* of $\mathbb{R}^{n \times m}$ (it is a copy of $\mathcal{O}(n)$), it has zero volume. Thus, as our second approximation, we discretize the conditional distribution by extending its support to the ambient space. Namely, we set $\hat{P}_{\mathbf{W}|\mathbf{B}}(w \mid \mathbf{B}) \propto \mathbb{1}[w \in \mathcal{W} \cap [\mathbf{B}]_{\delta/2}] \cdot 2^{-nm\delta}$, where $[\mathbf{B}]_{\delta/2}$ is the uniform $\delta/2$-expansion of $[\mathbf{B}]$:

$$[\mathbf{B}]_{\delta/2} = \{\mathbf{W} \in \mathbb{R}^{n \times m} \mid \exists \mathbf{Q} \in \mathcal{O}(n) : \|\mathbf{W} - \mathbf{Q}\mathbf{B}\|_\infty \leqslant \delta/2\}. \tag{16}$$

What is the size of $\mathcal{W} \cap [\mathbf{B}]_{\delta/2}$? Since $[\mathbf{B}]$ is a $n(n-1)/2$ dimensional subspace of $\mathbb{R}^{n \times m}$, for a large-enough precision $\delta$ we will have $|\mathcal{W} \cap [\mathbf{B}]_{\delta/2}| \approx 2^{-\delta \cdot n(n-1)/2}$. Though this approximation should be quite accurate, we do not expect equality in any practical situation; and the lack of this equality contributes to the numerical issues we describe in section 3.3.

Now, for large enough $\delta$, we have

$$\hat{P}_{\mathbf{W}|\mathbf{B}}(w \mid \mathbf{B}) \approx \mathbb{1}[w \in [\mathbf{B}]_{\delta/2}] \cdot 2^{-(nm-n(n-1)/2)\delta}.$$

Finally, assuming that $P_{\mathbf{B}}$ is uniform results in $P_{\mathbf{W}}$ being uniform as well, hence we have

$$\hat{P}_{\mathbf{W}}(w) = 2^{-nm\delta}.$$

Therefore, decoding $\mathbf{W} \mid \mathbf{B}$ saves approximately $-\log_2 \hat{P}_{\mathbf{W}|\mathbf{B}}(\mathbf{W} \mid \mathbf{B})$ bits and encoding it costs $-\log_2 \hat{P}_{\mathbf{W}}(\mathbf{W}) + n$ bits (where the $+n$ term comes from encoding the sign matrix $\sigma$), the total coding cost is

$$\approx \log_2 \frac{\hat{P}_{\mathbf{W}|\mathbf{B}}(\mathbf{W} \mid \mathbf{B})}{\hat{P}_{\mathbf{W}}(\mathbf{W})} + n \approx \log_2 \frac{2^{-(nm-n(n-1)/2)\delta}}{2^{-nm\delta}} + n = \frac{n(n-1)}{2}\delta + n \text{ bits},$$

which matches the coding cost of our proposed scheme.

## C  COMPARED WITH UNIVERSAL SOURCE CODING

As our proposed algorithm aims to reduce the storage and transmission cost, it is sensible to use a universal source coding algorithm to reduce the storage cost. However, we note that our proposed method is not directly comparable to these source coding algorithms. In fact, we can view our proposed algorithm as a pre-processing step before universal source coding. Concretely, we suggest to combine our proposed method with a source coding algorithm to form the following "bits-back" pipeline:

1. We obtain some network weights with rotational symmetries.

2. We use our method to eliminate the redundancies induced by the rotational symmetries in the weights.

3. We apply a universal source coding algorithm to the output of our algorithm.

In the main text, we only looked at the gains we get if we apply our method without any universal source coding. Hence, a concern arises: *do we retain significant gains if we run the pipeline we suggest above, compared to just running universal source coding without our method?*

Fortunately, the answer is positive. In particular, we compared the pipeline suggested above to universal source coding (we used ZIP in our experiment) on the OPT-2.7B model (slicing 30%). Using ZIP only, the compressed size comes to 3.97 GB while using our suggested bits-back pipeline, it reduces to 3.79 GB, and approximately 5% gain in storage size as before. While the exact gain my vary depending on the universal source coding algorithm, these models are large enough that the gains we report here should be fairly robust across different source coding algorithms.

There is an intuitive reason for retaining the gain even after source coding: the general-purpose universal source coding algorithm is unaware of the redundancies introduced by the rotational symmetry in the weights and, therefore, cannot utilize it to reduce storage size. From this perspective, the storage savings resulting from our bits-back method are "orthogonal" to the savings that result from source coding.

## D  ADDITIONAL EXPERIMENTS AND RESULTS

### D.1  RUNTIME ON CPU

Table 3: Encoding and decoding time on CPU for different models with varying slicing rates.

| Model Name | OPT-1.3B | | OPT-2.7B | | OPT-6.7B | | OPT-13B | |
|---|---|---|---|---|---|---|---|---|
| Slicing | 20% | 30% | 20% | 30% | 20% | 30% | 20% | 30% |
| Encoding time | 3.9 min | 3.5 min | 8 min | 6.5 min | 30 min | 25 min | 84 min | 68 min |
| Decoding time | 1.5 min | 1.5 min | 3.5 min | 2.5 min | 12 min | 10 min | 30 min | 24 min |

## D.2 INFLUENCE OF QUANTIZATION

In this paper, we apply our approach to weights saved in `float16`. Here, we provide a simple illustration of the influence of different precisions on the performance in Table 4. We apply simple linear quantization to reduce the precision to 11-15 bits and measure the compression rate with bits-back coding. We can see that as the precision decreases, the bits saved by our approach become smaller. This is because the size of the correction code increases to compensate for the error caused by lower precision. The influence of precision slightly varies for different model sizes and SliceGPT rates, but they share the same trend, and our proposed approach consistently delivers gains when the precision is larger than 12-13 bits. Also, note that we apply the simplest linear quantization to the model weights in this simple demonstration. A better quantization strategy and correction code can further improve the performance of our approach. We leave this investigation to further work.

Table 4: Influence of different precisions on the performance.

| Model / SliceGPT rate | | 16 bits | 15 bits | 14 bits | 13 bits | 12 bits | 11 bits |
|---|---|---|---|---|---|---|---|
| OPT-2.7B / 20% | w/o bits-back | | | -9.19% | | | |
| | w. bits-back | -13.84% | -13.29% | -13.04% | -12.10% | -9.83% | -3.92% |
| OPT-2.7B / 30% | w/o bits-back | | | -20.88% | | | |
| | w. bits-back | -24.43% | -24.09% | -23.86% | -23.12% | -21.60% | -17.20% |
| OPT-6.7B / 20% | w/o bits-back | | | -9.29% | | | |
| | w. bits-back | -14.07% | -13.29% | -12.88% | -11.58% | -8.01% | 0.00% |
| OPT-6.7B / 30% | w/o bits-back | | | -20.88% | | | |
| | w. bits-back | -24.84% | -24.27% | -23.91% | -22.96% | -20.38% | -14.14% |