# OpenReview forum: "Getting Free Bits Back from Rotational Symmetries in LLMs"
_ICLR.cc/2025/Conference — Submitted to ICLR 2025_

### Official Review · Reviewer_cA7j · 2024-11-02

**Soundness:** 2
**Presentation:** 2
**Contribution:** 2
**Rating:** 5
**Confidence:** 3

**Summary:**

This work proposes to apply a coding scheme to utilize symmetries made available by the SliceGPT method for training-free weight-only quantization. The resulting method achieves an additional 3-5% reduction in the total weight sizes for SliceGPT compressed models. The authors show that their resulting models do not diverge significantly from the original models when evaluated on tasks such as PIQA, WinoGrande, and HellaSwag.

**Strengths:**

1. The proposed method is training-free, making the result independent from the choice of calibration set.
2. The proposed method can be computed using only a CPU, without heavy computational requirements. This will make adoption easier.

**Weaknesses:**

1. The improvement of 3~5% appears small unless the method does not impose other overheads. However, there is no detailed analysis of how much memory the other components, such as the correction code, use.

2. Additionally, the work does not include any analysis of the overhead in terms of the time required for inference caused by applying additional computations to the model. Even if computing the rotation can be performed on CPU, there should be an analysis of the effect on inference latency when measured end-to-end.

**Questions:**

The beginning of Section 2 uses the word “delving” prominently. As the word “delve” is strongly associated with large language model outputs, we advise the authors to rephrase the sentence.

---

> ### Author Response · Authors · 2024-11-19
>
> Thank you for your valuable feedback, which helped us improve our manuscript. We are glad that you appreciate that our method is easy to adopt. Below, we reply to your concerns one by one.
>
> 1 **Weakness 1**:
>
> >The improvement of 3~5% appears small unless the method does not impose other overheads. However, there is no detailed analysis of how much memory the other components, such as the correction code, use.
>
> This a misunderstanding. The 3-5% bits already take the correction code into account, so the 3-5% savings are net. In fact, the length of the correction code is negligible.  We appreciate that you pointed out this potential confusion, and therefore, we have updated the caption of Table 1 to clarify this.
>
> 2. **Weakness 2**:
>
> > the work does not include any analysis of the overhead in terms of the time required for inference caused by applying additional computations to the model.
>
> Thank you for raising this question. We have updated our manuscript to include the results and discussion on the encoding and decoding time.
> Please refer to Table 2 and the runtime analysis paragraph on page 10. We also list Table 2 below for easier reference:
>
> | Model   Name   | OPT-1.3B | OPT-1.3B | OPT-2.7B | OPT-2.7B | OPT-6.7B | OPT-6.7B | OPT-13B | OPT-13B |
> |----------------|:--------:|:--------:|:--------:|:--------:|:--------:|:--------:|:-------:|:-------:|
> | Slicing        |    20%   |    30%   |    20%   |    30%   |    20%   |    30%   |   20%   |   30%   |
> | Encoding  time |   15 s   |   13 s   |   30 s   |   24 s   |  2.5 min |  1.7 min | 6.5 min | 4.1 min |
> | Decoding  time |    6 s   |    5 s   |   14 s   |   11 s   |  1.2 min |   45 s   | 2.5 min |  2 min  |
>
>
> As we can see, our algorithm only takes seconds to minutes on a consumer-grade GPU. We also provide the runtime on CPU in Appendix C.1. Also, as we discussed in lines 501-511, we can even parallelize this process to accelerate the execution further.
>
> Additionally, we note that our approach aims to reduce the storage space and transmission cost. Therefore, we will only run the encoding algorithm when saving the model and decoding when loading the model. Once the model is decoded and loaded into memory, the inference time will be identical to the vanilla Transformer.
> Therefore, when measured end-to-end, our approach only increases the whole procedure by a few seconds/minutes. Therefore, this is not a big obstacle for practical usage.
>
> 3.
>
> >The beginning of Section 2 uses the word “delving” prominently. As the word “delve” is strongly associated with large language model outputs, we advise the authors to rephrase the sentence.
>
> Thank you for your suggestions! We have rephrased this sentence.
>
>
>
> *Thank you once again for taking the time to review our work and our response. We believe we have addressed your questions.
>  We are happy to discuss any further concerns you might have. However, If you find our clarifications satisfactory, we kindly invite you to raise the score.*

---

> ### Author Response · Authors · 2024-11-21
> **Thank you for your review! Please consider our response**
>
> Thank you again for the effort you put into our work. As only a few working days are left in the discussion period, we would like to ask if our response has satisfied your concerns. If so, we kindly invite you to consider raising the score. If any concerns remain, we are happy to discuss them further here.

---

> ### Comment · Reviewer_cA7j · 2024-11-22
>
> I thank the authors for their detailed response. It has helped clarify the scope and motivation of the work.
> However, if the aim of the paper is to reduce disk storage, I believe that much more extensive comparisons should be made with compression methods such as Zstandard and byte shuffling, which also have the advantage of being lossless. Lossy algorithms for floating-point data compression also exist.
> In general, the model compression algorithms are designed to reduce the amount of memory in the GPU, not disk storage. This is because GPU memory is a scarcer resource, and reducing the required GPU memory allows models to run using fewer GPUs, which reduces costs. In addition, GPU memory must be accessed once per decoding step of the model, whereas storage needs only be accessed once at the very beginning when the model is being loaded to RAM.
> Because of this dynamic, I am curious whether there is a motivating use case in mind. For example, if many user-defined models need to be saved and transferred frequently, there may be a case to reduce the storage requirements, even at the cost of introducing lossy compression. However, even in this case, a rigorous comparison against both lossless and lossy techniques for model weight compression should be conducted for a fair comparison.
> Due to the difficulty of understanding the motivation of the work, I will keep my score.

---

> > ### Author Response · Authors · 2024-11-27
> > **Thank you for your review and discussion! Please consider our response**
> >
> > Thank you for the effort you put into reviewing and discussing. We would like to ask if you could look at our last response and kindly invite you to increase your score if our response clarifies our method and addresses your concerns. If any concerns remain, we are happy to continue discussing them further.

---

> > > ### Comment · Reviewer_cA7j · 2024-11-28
> > >
> > > We thank the authors for their detailed reply.
> > > However, we maintain that the gain from reducing storage memory does not justify introducing a lossy quantization algorithm.
> > > Several lossless alternatives which serve a similar purpose exist, for example, the byteshuffle and bitshuffle filters available in the Zarr library. Applying these filters improves the compressibility of input data, even for lower compression levels of standard compression algorithms such as GZip and ZStandard. For example, applying GZip level 1 with the byteshuffle filter often produces a better result than applying GZip level 6.
> > > Moreover, while network bandwidth is sometimes a bottleneck, this problem can often be mitigated by caching frequently used components in faster storage. For example, small weights such as LoRA can be kept in RAM instead of being transferred.
> > > As mentioned in our previous reply, the costs of a large storage footprint are only paid once at the first step, while the costs of a large memory footprint in the GPU are paid every time the model executes. While a lossy algorithm may be justifiable to reduce the latter problem, it is not justifiable to reduce the former.

---

> ### Author Response · Authors · 2024-11-22
>
> Thank you for your reply; you make excellent points, and they made us see our work from a new angle.
>
> > I believe that much more extensive comparisons should be made with compression methods such as Zstandard and byte shuffling, which also have the advantage of being lossless.
>
> You are correct that if we are storing the weights on the disk, then it is sensible to use a universal source coding algorithm such as the ones you mention to ensure we save as much space as possible.
>
> However, our proposed method is not directly comparable to these source coding algorithms. In fact, we can view our proposed algorithm as a pre-processing step before universal source coding. To be completely clear, we suggest that we combine our proposed method with a source coding algorithm to form the following "bits-back" pipeline:
> 1. We obtain some network weights with rotational symmetries.
> 2. We use our method to eliminate the redundancies induced by the rotational symmetries in the weights.
> 3. We apply a universal source coding algorithm (e.g. Zstandard) to the output of our algorithm.
> 4. We save the output of this pipeline to the disk.
>
> In our paper, we only looked at the gains we get if we apply our method only, without any universal source coding. Hence, an important concern arises: do we retain significant gains if we run the pipeline we suggest above, compared to just running universal source coding without our method?
>
> Fortunately, the answer is positive. In particular, we compared the pipeline suggested above to universal source coding (we used ZIP in our experiment) on the OPT-2.7B model. Using ZIP only, **the compressed size comes to 3.97 GB while using our suggested bits-back pipeline, it reduces to 3.79 GB, and approximately 5% gain in storage size as before**. Note, that while the exact gain my vary depending on the universal source coding algorithm we use, these models are large enough that the gains we report here should be fairly robust across different source coding algorithms.
>
> There is an intuitive reason for retaining the gain even after source coding: ZIP (or any other general-purpose universal source code) is unaware of the redundancies introduced by the rotational symmetry in the weights and, therefore, cannot utilise it to reduce storage size. From this perspective, **the storage savings resulting from our bits-back method are "orthogonal" to the savings that result from source coding.** Essentially, to the best of our knowledge, we are the first to address these structural redundancies in neural networks through bits-back coding. As such, our approach is not directly comparable to other compression algorithms that do not account for this specific type of redundancy.
>
> Once again, thank you for this excellent point; we have also updated our paper to include the above discussion in the appendix!
>
> **Regarding your second concern about the motivation of our approach**, we want to highlight 3 points:
> 1. We agree GPU is scarcer compared to disk. However, bandwidth is also another bottleneck. Therefore, *reducing storage not only reduces disk costs but also reduces transmission costs and speeds up weights sharing*.
> 2. One potential application that reflects point 1 is applying Federated Learning for LLMs, or for LoRAs. In these applications, the client needs to frequently send messages of weights/gradients to the server. Therefore, our approach can gain potential application in these areas.
> 3. we also want to note that, our approach can save 3-5% bits mostly for free and there is almost no drawback for applying our approach given its minimal runtime overhead and negligible influence on performance. Therefore, *it is always beneficial to apply this approach in the cases where we need to store and transmit weights*.
>
> We thank you again for your detailed reply. If you find our further clarifications satisfactory, we kindly invite you to consider raising the score.

---

> ### Author Response · Authors · 2024-12-01
>
> Thank you for your reply. However, given your comments, we believe there are three fundamental facts that we haven’t communicated clearly enough, and thus you appear to have misunderstood:
>
> 1. **Our method is NOT a lossy quantization algorithm.** Rather, our method is a lossless operation (up to floating-point precision) that we can apply before saving the weights.
> 2. **Our method is NOT comparable with the methods you suggest.** The methods you list, such as byteshuffle, are general preprocessing methods to improve the compressibility of model weights, and GZip and ZStandard are universal source codes. On the other hand, our method’s sole purpose is to **eliminate the description inefficiency induced by rotational symmetries in the weights**. This incomparability also means that **we can freely combine our method with those you mention**. Our previous response demonstrated how our method retains the ~5% gain in storage space when we combine it with Zip. We have now run some experiments where we apply Zstd *with and without bshuffling* as well as *with and without applying our method* on OPT-2.7 model weights:
>
> |             | Zstd, autoshuffle | Zstd, noshuffle |
> |-------------|-------------------|-----------------|
> | **without** our method | 3.48 GB           | 3.78 GB         |
> | **with** our method | 3.33 GB           | 3.60 GB         |
>
> As you can see, **our algorithm still provides ~5% gain, regardless of whether we use shuffling, or not.**
>
> 3. **We are NOT proposing to apply our method to reduce GPU usage**, only storage. Reducing GPU cost is out of the scope of design intent, as outlined in our manuscript and previous discussion.
>
>
> To summarise, our method aims to reduce the storage cost of network weights, e.g., LLMs, at a negligible additional computational cost. **Our method eliminates a redundancy that no other method can/does** by leveraging the extra knowledge that there are rotational symmetries in the model weights.
> As such, we believe that our method is a significant and valuable addition to the LLM and the broader model compression literature.
> Thank you for the time you have invested in reviewing our paper and actively engaging in discussion with us. We believe we have addressed your concerns and thus kindly invite you once more to raise your score.

---

### Official Review · Reviewer_r3xU · 2024-11-03

**Soundness:** 3
**Presentation:** 3
**Contribution:** 3
**Rating:** 8
**Confidence:** 2

**Summary:**

The paper presents a novel training-free compression technique of large language models that exploits rotational symmetries in the weight space. It uses bits-back coding, a compression strategy that takes advantage of these rotational symmetries to compress Transformer models by about three to five percent while impacting the model's perplexity in a negligible way. The method was tested on SliceGPT-pruned Transformers, namely the OPT and Llama-2.

**Strengths:**

- The paper presents bits-back coding used on neural network models, mainly focusing on enlarging language compression.
- The proposed method is computationally feasible since it runs without retraining.
- The paper's novel technique is evaluated on models, such as OPT and Llama-2, demonstrating performance metrics are not significantly affected in terms of perplexities drop.

**Weaknesses:**

-  This approach is inherently SliceGPT pruning and Transformer-specific architecture, which may also limit its use to other neural networks or pruning techniques.
- The methodology relies only on Transformer architectures, so applicability to lighter models suited to edge devices could be considered.

**Questions:**

- What is the prospective effectiveness of the method when it comes to implementation on the models with precision format lower than float16?
- Is it possible to apply the bits-back coding method to the architectures that are not transformers or the architectures compressed with different methods?

---

> ### Author Response · Authors · 2024-11-19
>
> Thank you for your valuable feedback. We are encouraged by the positive comments around soundness and novelty. We address your concerns in the following.
>
> 1. **Weakness 1 & Question 2**
>
> >This approach is inherently SliceGPT pruning and Transformer-specific architecture, which may also limit its use to other neural networks or pruning techniques.
>
> >Is it possible to apply the bits-back coding method to the architectures that are not transformers or the architectures compressed with different methods?
>
> We agree that the algorithm described in our manuscript focuses on SilceGPT. However, we emphasize that the concept of bits-back coding applied to neural networks is general and can be extended to other architecture exhibiting symmetries. One of our key contributions, as noted in the conclusion section, is establishing a connection between bits-back coding and networks with symmetrical properties. Below, we provide two examples where our algorithm (or its variations) can be employed to achieve free bits-back.
> We have also included these discussions in the Conclusion, Limitations, and Future Directions Section in our updated manuscript.
>
> (1) Applying our approach to encoding LoRA modulation or weights decomposed in LoRA-style. Specifically, LoRA approximates a matrix $M$ by $M=BA$. We can see applying $Q$ to $B$ and $Q^T$ to $A$ will leave M invariant. Therefore, our proposed approach can be seamlessly applied in this context.
>
> (2) Neural networks with permutation symmetry. It is well known that in many architectures, such as standard MLPs, permuting the hidden units in one layer and applying the reverse permutation to the subsequent layer leaves the output unchanged.
> We can modify our approach to apply bits-back coding in this case.
> Specifically, we can define a canonical order for hidden units based on a predefined criterion (e.g., sorting the weights or biases corresponding to each unit in descending order). During encoding, a permutation can be randomly selected by decoding bits from the current bitstream. During decoding, this permutation can be easily recovered by rearranging the hidden units back to their canonical order.
> In fact,  it is a standard fact that permutation matrices are orthogonal (i.e., they can be thought of as rotations). Therefore, they can also be seen as a special case of our approach.
>  However, the gain here is smaller than for the rotational symmetry case, as the permutation invariance of the hidden units introduces significantly less redundancy compared to rotations in general.
>
> 2. **Weakness 2**
>
> > Applicability to lighter models suited to edge devices could be considered.
>
> Thank you for your valuable suggestion; applying our method to smaller models could be quite impactful! We only considered SliceGPT in our paper, because its parameters exhibit a very clear rotational invariance that we could exploit. If the reviewer can recommend other, smaller architectures with well-known parameter symmetries, we would be very grateful and interested to hear!
>
>
> 3. **Question**
>
> >What is the prospective effectiveness of the method when it comes to implementation on the models with precision format lower than float16?
>
> Quantization can impact the performance of our approach. In our algorithm, the rotated matrix $WQ$ is encoded into the bitstream. During decoding, we use eigenvalue decomposition to recover $Q$ from $WQ$. If $WQ$ is quantized to a low precision, the quantized version may differ from the original $WQ$. As a result, performing eigenvalue decomposition on the quantized matrix may produce a recovered $\hat{Q}$ that has a larger difference compared to the original $Q$.
>
> To illustrate this impact, we provide an extra experiment in Appendix C.2 in our updated manuscript. We apply simple linear quantization to reduce the precision to 11-15 bits and measure the compression rate with bits-back coding. We can see that as the precision decreases, the bits saved by our approach become smaller. This is because the size of the correction code increases to compensate for the error caused by lower precision.
> Our proposed approach consistently delivers gains when the precision is larger than 12-13 bits. We also want to note that we apply the simplest linear quantization in this simple demonstration. A better quantization strategy and correction code can further improve the performance of our approach. Designing a method that would allow us to quantize the weights below 12-bit precision is an interesting avenue for future research.
>
>
> *Thank you once again for taking the time to review our work and our response. We are happy to discuss any further questions you might have. However, should we have addressed your concerns, we kindly invite you to raise the score.*

---

> ### Author Response · Authors · 2024-11-21
> **Thank you for your review! Please consider our reply**
>
> Thank you once again for the effort you put into reviewing our paper. As only a few working days are left in the discussion period, we would like to ask if our response has satisfied your concerns. If so, we kindly invite you to consider raising the score. If any concerns remain, we are happy to discuss them further here.

---

> > ### Comment · Reviewer_r3xU · 2024-11-22
> > **Update Reviews**
> >
> > Dear Authors,
> >
> > Thank you for addressing my questions regarding the effectiveness of your approach with a lower precision format and the generalization of your approach to other neural network architectures. I thus increased my rating score.
> >
> > Best regards.

---

### Official Review · Reviewer_Li28 · 2024-11-04

**Soundness:** 3
**Presentation:** 3
**Contribution:** 3
**Rating:** 5
**Confidence:** 3

**Summary:**

This paper proposes the application of the bits-back algorithm to reduce the overhead of additional matrices introduced by pruning large language models (LLMs) using SliceGPT. In the SliceGPT method, an additional matrix is introduced for the rotation matrix, which helps in maintaining accuracy but results in additional computational overhead, thereby acting as a compression overhead. To address this issue, we propose an algorithm that encodes/decodes the rotation matrix using the bits-back algorithm, demonstrating that the rotation matrix can be computed solely through the decoding process during inference. Our proposed method shows an additional 3-5% improvement in compression efficiency compared to the practical compression rate of SliceGPT.

**Strengths:**

* The paper proposes a method to compress the rotation matrix Q introduced by SliceGPT using the bits-back algorithm, effectively reducing the parameter overhead.
* It demonstrates that the rotation matrix Q can be encoded and decoded using the bits-back algorithm without requiring a calibration set, relying solely on the weight matrix.
* The study shows that while the actual compression rate of SliceGPT with the rotation matrix Q is approximately 9%, the proposed method can achieve a closer-to-expected compression rate of 13%. It also demonstrates that applying the proposed encoding method to the rotation matrix Q in models such as OPT and LLaMA2-7B does not result in significant differences in Commonsense Reasoning (CSR) performance.

**Weaknesses:**

* The paper lacks sufficient analysis and experimentation regarding the practical impact on latency and throughput during inference when decoding the rotation matrix Q using the proposed method.
* The proposed method is somewhat limited in scope, as it can only be applied after the implementation of SliceGPT, thereby restricting its applicability.
* The actual benefits of encoding the rotation matrix Q in terms of inference latency and throughput might be minimal. It is likely that during the prefill stage, the additional decoding step for the rotation matrix Q could result in higher inference latency and lower throughput compared to SliceGPT alone.

**Questions:**

* How does the proposed method perform in terms of inference latency and throughput gains compared to SliceGPT when applied on actual hardware like GPUs? If the effectiveness of this aspect is demonstrated, I would be inclined to increase my rating.

---

> ### Author Response · Authors · 2024-11-19
>
> Thank you for your insightful review and questions, which have helped us improve our manuscript.  We now reply to the stated weaknesses and questions.
>
> 1. **weaknesses 1 & 3, and question** (regarding impact on latency and throughput):
> > The paper lacks sufficient analysis and experimentation regarding the practical impact on latency and throughput during inference when decoding the rotation matrix Q using the proposed method. The actual benefits of encoding the rotation matrix Q in terms of inference latency and throughput might be minimal. How does the proposed method perform in terms of inference latency and throughput gains compared to SliceGPT when applied on actual hardware like GPUs?
>
> Thank you for raising this concern. While our approach requires additional time for decoding, there is no influence on the latency and throughput during inference:
>
> (1) Our approach aims to reduce storage space and transmission costs. Therefore, we will only run the encoding algorithm when saving the model and decoding when loading the model. Once the model is decoded and loaded into memory, there is no overhead on the influence of the inference time and throughput.
>
> (2) We measured the encoding and decoding time and updated our manuscript to include the results and discussion. Please refer to Table 2 and the runtime analysis paragraph on page 10.  We also show Table 2 here for easy reference:
>
> | Model   Name   | OPT-1.3B | OPT-1.3B | OPT-2.7B | OPT-2.7B | OPT-6.7B | OPT-6.7B | OPT-13B | OPT-13B |
> |----------------|:--------:|:--------:|:--------:|:--------:|:--------:|:--------:|:-------:|:-------:|
> | Slicing        |    20%   |    30%   |    20%   |    30%   |    20%   |    30%   |   20%   |   30%   |
> | Encoding  time |   15 s   |   13 s   |   30 s   |   24 s   |  2.5 min |  1.7 min | 6.5 min | 4.1 min |
> | Decoding  time |    6 s   |    5 s   |   14 s   |   11 s   |  1.2 min |   45 s   | 2.5 min |  2 min  |
>
> We can see that decoding the entire network is actually fast on the consumer-grade GPU. Therefore, even if we measure the inference time end-to-end, the influence of our approach on runtime is minimal.
>
> (3) Additionally, we did not optimize our implementation for the decoding runtime, and there are ways to improve it. We discuss one such possibility in lines 501-511 in our updated manuscript: parallelizing the encoding and decoding to accelerate the execution.
>
> 2. **weakness 2:**
> >The proposed method is somewhat limited in scope, as it can only be applied after the implementation of SliceGPT, thereby restricting its applicability.
>
> We agree that our approach focuses on SilceGPT. However, we emphasize that the concept of bits-back coding applied to neural networks is general and can be extended to any architecture exhibiting symmetries. One of our key contributions, as noted in the conclusion section, is establishing a connection between bits-back coding and statistical models that exhibit invariance under rotations and, more generally, under the action of the elements of a symmetry group. Below, we provide two examples where our algorithm (or its variations) can be employed to achieve free bits-back. We have also included these discussions in the Conclusion, Limitations, and Future Directions Section in our updated manuscript.
>
> (1) Applying our approach to encoding LoRA modulation or weights decomposed in LoRA-style. Specifically, LoRA approximates a matrix $M$ by $M=AB$. We can see applying $Q$ to $A$ (as $AQ$) and $Q^T$ to $B$ (as $Q^TB$) will leave M invariant. Therefore, our proposed approach can be seamlessly applied in this context.
>
> (2) Neural networks with permutation symmetry. It is well known that in many architectures, such as standard MLPs, permuting the hidden units in one layer and applying the reverse permutation to the subsequent layer leaves the output unchanged.
> We can modify our approach to apply bits-back coding in this case.
> Specifically, we can define a canonical order for hidden units based on a predefined criterion (e.g., sorting the weights or biases corresponding to each unit in descending order). During encoding, a permutation can be randomly selected by decoding bits from the current bitstream. During decoding, this permutation can be easily recovered by rearranging the hidden units back to their canonical order. In fact, it is a standard fact that permutation matrices are orthogonal (i.e., they can be thought of as rotations). Therefore, they can also be seen as a special case of our approach. However, the gain here is smaller than for the rotational symmetry case, as the permutation invariance of the hidden units introduces significantly less redundancy compared to rotations in general.
>
> *Thank you once again for taking the time to review our work and our response. We are happy to discuss any further questions you might have. However, should we have addressed your concerns, we kindly invite you to raise the score.*

---

> ### Author Response · Authors · 2024-11-21
> **Thank you for your review! Please consider our response**
>
> Thank you again for the effort and time you put into our paper. As only a few working days are left in the discussion, we would like to ask if our response has satisfied your concerns. If so, we kindly invite you to consider raising the score. If any concerns remain, we are happy to discuss them further here.

---

> > ### Comment · Reviewer_Li28 · 2024-11-25
> > **Response to the authors**
> >
> > Thank you for the detailed response. Most of my questions have been clarified. However, I believe that the author's method can provide significant latency and throughput benefits if some accuracy degradation is acceptable. In the generation phase of LLM inference, the batch size is usually small, and the proposed method can benefit from the memory-bound nature of LLM inference. I think that if the proposed method can focus on optimizing the decoding algorithm, it can improve latency and throughput by sacrificing additional computational cost and reducing significant memory communication cost. For this reason, I would like to keep the original score unless further explanation is provided.

---

> > > ### Author Response · Authors · 2024-11-25
> > >
> > > Thank you for your reply; we now address your comments below.
> > >
> > > 1. > However, I believe that the author’s method can provide significant latency and throughput benefits if some accuracy degradation is acceptable.
> > >
> > > We are unsure about what you mean by this comment for two reasons:
> > >
> > > a. As we have noted in our rebuttal already, our proposed approach has **no influence on the inference latency and throughput** of the LLM. It only slightly affects the time it takes to load the model weights from the hard drive into the memory.
> > >
> > > b. Our paper demonstrated that **our method does not lead to accuracy degradation.** In Table 1 of our paper, we show that there is no statistically significant accuracy degradation after we apply our compression method: roughly half of the time, the model accuracy slightly decreases, while half of the time, it increases after we apply our method.
> > > Therefore, **there is no inherent trade-off between accuracy degradation and latency or throughput in our approach.**
> > >
> > > To sum up, our contribution focuses on connecting bits-back coding with model compression for efficient storage and transmission. This is a standard, general-purpose model compression task. For its intended purpose, our current approach is already complete, efficient, and practical, as we have demonstrated in our paper and prior replies. Our method does not influence latency and has no trade-off between accuracy degradation and latency or throughput.
> > >
> > > 2. > In the generation phase of LLM inference, the batch size is usually small, and the proposed method can benefit from the memory-bound nature of LLM inference.
> > >
> > > This is an interesting suggestion, but we do not quite understand it. We note that our contribution focuses on connecting bits-back coding with model compression for efficient storage and transmission. The scenario you describe seems to diverge from the intended design of our work. Could you please further clarify how this connects with our proposed approach?
> > >
> > > 3. >  I think that if the proposed method can focus on optimizing the decoding algorithm, it can improve latency and throughput by sacrificing additional computational cost and reducing significant memory communication cost.
> > >
> > > To clarify, our proposed method does not influence latency and throughput. Rather, it aims to reduce the space needed to store the model on the hard drive and to reduce the transmission cost of sharing models. We wonder whether our usage of the term “*decoding*” might be causing this misunderstanding. We use the term from the source coding and information theory perspective, referring to reading and processing the compressed weights from the hard drive and loading them into memory. You can understand this term as "*decompression*", rather than the process in LLM inference. For a more elaborate discussion on this topic, please refer to [our global response to the reviewers](https://openreview.net/forum?id=B8aHIDSi7E&noteId=hPIqJTx60Y).
> > >
> > > We thank you once again for your effort to review our paper. Given that not much time is left in the discussion period, we would appreciate it if you could clarify your comments as soon as possible. On the other hand, if we have addressed your concerns and clarified our position, we kindly invite you to reconsider raising your score.

---

> > > ### Author Response · Authors · 2024-11-27
> > > **Thank you for your review and discussion! Please consider our response**
> > >
> > > Thank you for the effort you put into reviewing and discussing. We would like to ask if you could look at our last response and kindly invite you to increase your score if our response clarifies our method and addresses your concerns. If any concerns remain, we are happy to continue discussing them further.

---

### Official Review · Reviewer_kYZS · 2024-11-06

**Soundness:** 3
**Presentation:** 2
**Contribution:** 3
**Rating:** 6
**Confidence:** 3

**Summary:**

In this paper, the authors highlight that the rotational symmetries of SliceGPT introduce redundancies. Based on SliceGPT, they propose further compressing the weights by the bits-back coding algorithm. Specifically, rather than treating all weight configurations as unique, they encode weights up to an equivalence class defined by rotations, enabling smaller memory requirements. They conducted experiments on several benchmarks with multiple models. The results verify the effectiveness of their proposed method.

**Strengths**
1. The proposed method is novel and insightful
2. The proposed method is well-motivated and has the potential to make a broader impact.
3. The experiment results are promising.

**Weaknesses**
1. The writing is not self-contained. Specifically, the paper relies heavily on bits-back coding. However, they do not properly connect SliceGPT with the previously proposed bits-back coding. It is hard to understand the actual algorithm.
2. It is questionable if the method can be applied in the real world given the compression/decompression and matrix decomposition procedures involved. The run speed of this method could be slower than that of the vanilla Transformer model.

In summary, the paper is insightful and well-motivated. However, the writing is not self-contained and the overhead could hinder the real-world application. As a result, I recommend a weak acceptance.

**Strengths:**

1. The proposed method is novel and insightful
2. The proposed method is well-motivated and has the potential to make a broader impact.
3. The experiment results are promising.

**Weaknesses:**

1. The writing is not self-contained. Specifically, the paper relies heavily on bits-back coding. However, they do not properly connect SliceGPT with the previously proposed bits-back coding. It is hard to understand the actual algorithm.
2. It is questionable if the method can be applied in the real world given the compression/decompression and matrix decomposition procedures involved. The run speed of this method could be slower than that of the vanilla Transformer model.

**Questions:**

See the weaknesses.

---

> ### Author Response · Authors · 2024-11-19
>
> Thank you for your detailed review and constructive questions. We are glad that you found our paper novel, insightful, and to have the potential to have a broader impact. We now reply to your two questions below.
>
> 1. **Question 1**:
> >the paper relies heavily on bits-back coding. However, the paper does not properly connect SliceGPT with the previously proposed bits-back coding. It is hard to understand the actual algorithm.
>
> This seems to be a misunderstanding. There is no a priori connection between SliceGPT and bits-back coding. In fact, making this connection is one of our contributions.
>
> Bits-back coding is a data compression algorithm that works in cases where multiple choices are acceptable or equally good. In our case, as we explained in Remark 3.1, SliceGPT introduces symmetries into Transformers. Specifically, different weights, up to any random rotation, are equally good. Therefore, we can connect bits-back coding to SliceGPT. We then explain the benefit of this connection through an informal calculation of the codelength.
>
> We appreciate this concern and agree that this connection does not seem crystal clear at first glance. Therefore, we have added a more detailed motivation from Lines 140-144.
>
> 2. **Question 2**:
> >It is questionable if the method can be applied in the real world … the run speed of this method could be slower than that of the vanilla Transformer model.
>
> Thank you for raising this question. However, this is not a concern:
>
> (1) Our approach aims to reduce the storage space and transmission cost. Therefore, we will only run the encoding algorithm when saving the model and decoding when loading the model. Once the model is decoded and loaded into memory, the inference time will be identical to the vanilla SliceGPT.
>
> (2) We measured the encoding and decoding time and updated our manuscript to include the results and discussion. We list Table 2 in the updated manuscript for an easier reference:
>
> | Model   Name   | OPT-1.3B | OPT-1.3B | OPT-2.7B | OPT-2.7B | OPT-6.7B | OPT-6.7B | OPT-13B | OPT-13B |
> |----------------|:--------:|:--------:|:--------:|:--------:|:--------:|:--------:|:-------:|:-------:|
> | Slicing        |    20%   |    30%   |    20%   |    30%   |    20%   |    30%   |   20%   |   30%   |
> | Encoding  time |   15 s   |   13 s   |   30 s   |   24 s   |  2.5 min |  1.7 min | 6.5 min | 4.1 min |
> | Decoding  time |    6 s   |    5 s   |   14 s   |   11 s   |  1.2 min |   45 s   | 2.5 min |  2 min  |
>
>
> For a more detailed discussion, please refer to the runtime analysis paragraph on page 10. As we can see, our algorithm only takes seconds to minutes on a consumer-grade GPU. Also, as we discussed in lines 501-511, we can even parallelize this process to accelerate the execution further.
>
> Therefore, the runtime does not pose an obstacle to deploying our algorithm in real-world applications. Moreover, our algorithm can also be executed on a CPU, further broadening its applicability. We also include the encoding and decoding times for CPU execution in Appendix C.1 in the updated manuscript.
>
> *Thank you once again for taking the time to review our work and our response. We believe we have addressed your questions. If you find our clarifications satisfactory, we kindly invite you to consider raising your score.*

---

> > ### Author Response · Authors · 2024-11-27
> > **Thank you for your review! Please consider our response**
> >
> > Thank you for the effort you put into reviewing. We would like to ask if you could look at our response and kindly invite you to consider increasing your score if our response addresses your concerns. We are happy to continue discussing them further if any concerns remain.

---

> > > ### Comment · Reviewer_kYZS · 2024-12-03
> > >
> > > > Weakness 1
> > >
> > > This does not seem to be a misunderstanding. I was aware that there is no prior connection, and I believe that the authors need to "properly connect SliceGPT with the previously proposed bits-back coding". Specifically, the authors need to illustrate in detail what these techniques are and why SliceGPT could fit in the framework of bits-back coding. So far, I have not found this addressed yet.
> > >
> > > > Weakness 2
> > >
> > > Thank you for the information. The question is now clear.
> > >
> > > In general, the authors partially addressed my concerns. Since I have already suggested an acceptance, I maintain my original score.

---

> > > > ### Author Response · Authors · 2024-12-03
> > > >
> > > > Thank you for your great suggestions. We apologize for misunderstanding your previous concerns. Since we cannot update our PDF at this stage, we commit to including a clearer explanation connecting bits-back and sliceGPT in the camera-ready version to make our paper easier to follow.

---

> ### Author Response · Authors · 2024-11-21
> **Thank you for your review! Please consider our response**
>
> Thank you once again for the effort you put into reviewing our paper. As only a few working days are left in the discussion period, we would like to ask if our response has satisfied your concerns. If so, we kindly invite you to consider raising the score. If any concerns remain, we are happy to discuss them further here.

---

### Author Response · Authors · 2024-11-19
**Global Response to all reviewers**

We extend our gratitude to all the reviewers for their detailed and comprehensive reviews and for the time spent reviewing our manuscript. We are pleased that the reviewers acknowledged the novelty, insightfulness, and applicability of our approach. We have carefully addressed their concerns in our responses, and have modified our manuscript accordingly (modifications are highlighted in red).

Most of the reviewers raised two common concerns. Therefore,  we provide a concise summary of our responses in this global response.

1. **Regarding the runtime and the influence of our approach on inference latency:**

We highlight that our approach aims to reduce the storage space and transmission cost. Therefore, we will only run the encoding algorithm when saving the model and decoding when loading the model. Our approach will not influence the inference time once the model is decoded and loaded into memory.

To evaluate the influence on the model saving and loading time, we have updated our manuscript to include encoding and decoding time in Table 2 and discussion in the runtime analysis paragraph on page 10. Our algorithm only adds seconds to minutes to the loading time on a consumer-grade GPU. We also provide the runtime on the CPU in Appendix C.1.



2. **Our proposed method is specific to SliceGPT:**

We emphasize that the concept of bits-back coding applied to neural networks is general and can be extended to other architectures exhibiting symmetries. One of our key contributions, as noted in the conclusion section, is establishing a connection between bits-back coding and networks with symmetrical properties. In our reply to the reviewers, we provide two examples (LoRA and networks with Permutation symmetry) where our algorithm (or its variations) can be employed to achieve free bits back. We have updated our manuscript to include a discussion on this at the end.

---

> ### Author Response · Authors · 2024-11-22
>
> 3. **Review cA7j raised a good question: as our aim is to reduce the storage and transmit cost, why we do not just use some source coding algorithm?**
> We believe that question allows us to see our work from a new angle and, hence, also respond to it in the global response:
>
> To answer this question, we highlight that we should not directly compare our proposed method to these source coding algorithms. In fact, we can view our proposed algorithm as a pre-processing step before universal source coding. To be completely clear, we suggest that we combine our proposed method with a source coding algorithm as follows:
>
> 1. We obtain some network weights with rotational symmetries.
>
> 2. We use our method to eliminate the redundancies induced by the rotational symmetries in the weights.
>
> 3. We apply a universal source coding algorithm (e.g. Zstandard) to the output of our algorithm.
>
> 4. We save the output of this pipeline to the disk.
>
> In our paper, we looked at the gains we get if we apply our method only, without any universal source coding. In this response, we showcased that we still retain significant gains if we run the pipeline we suggest above, compared to just running universal source coding without our method.
> In particular, we compared the pipeline suggested above to universal source coding (we used ZIP in our experiment) on the OPT-2.7B model. Using ZIP only, the compressed size comes to 3.97 GB while using our suggested bits-back pipeline, it reduces to 3.79 GB, and approximately 5% gain in storage size as before.
> There is an intuitive reason for retaining the gain even after source coding: ZIP (or any other general-purpose universal source code) is unaware of the redundancies introduced by the rotational symmetry in the weights and, therefore, cannot utilize it to reduce space. **From this perspective, the storage savings resulting from our bits-back method are “orthogonal” to the savings that result from source coding.**

---

### Meta-Review · Area_Chair_hrx2 · 2024-12-15

**Metareview:**

This paper proposes a method to compress rotationally symmetric weight matrices. Due to their rotational symmetry, the encoding of these weight matrices can be redundant. To address this, the authors propose the bits-back coding algorithm for compression.
Main strengths:
- The paper makes an interesting algorithmic contribution.

Main weaknesses:
- The proposed method has only been benchmarked on SliceGPT pruning and Transformers.
- The achieved savings are limited.

It will help if the authors take concrete steps to demonstrate broader applicability of the method.

**Additional Comments On Reviewer Discussion:**

Some points have been clarified by the authors during the rebuttal:
- The coding scheme is primarily designed to reduce storage and transmission costs and is therefore not significantly related to latency in model predictions.
- The coding scheme uses rotational symmetry to save bits, which complements typical compression methods.

While these points are clear, I believe the paper could be further improved by demonstrating broader applications beyond a single pruning method. It would also be useful to explore general rotational symmetries in weight matrices and evaluate the potential savings achieved through this approach.

---

### Decision · Program_Chairs · 2025-01-22

Reject